# Developments in Synthesis and Potential Electronic and Magnetic Applications of Pristine and Doped Graphynes

**DOI:** 10.3390/nano11092268

**Published:** 2021-08-31

**Authors:** Gisya Abdi, Abdolhamid Alizadeh, Wojciech Grochala, Andrzej Szczurek

**Affiliations:** 1Centre of New Technologies, University of Warsaw, S. Banacha 2c, 02-097 Warsaw, Poland; abdi_gisya@yahoo.com (G.A.); w.grochala@cent.uw.edu.pl (W.G.); 2Academic Centre for Materials and Nanotechnology, AGH University of Science and Technology, al. A. Mickiewicza 30, 30-059 Krakow, Poland; 3Department of Organic Chemistry, Faculty of Physics and Chemistry, Alzahra University, Tehran 1993893973, Iran; ahalizadeh2@hotmail.com

**Keywords:** graphyne-like materials, synthesis and doping, electronic and magnetic properties, electronic transport, photodetectors

## Abstract

Doping and its consequences on the electronic features, optoelectronic features, and magnetism of graphynes (GYs) are reviewed in this work. First, synthetic strategies that consider numerous chemically and dimensionally different structures are discussed. Simultaneous or subsequent doping with heteroatoms, controlling dimensions, applying strain, and applying external electric fields can serve as effective ways to modulate the band structure of these new sp^2^/sp allotropes of carbon. The fundamental band gap is crucially dependent on morphology, with low dimensional GYs displaying a broader band gap than their bulk counterparts. Accurately chosen precursors and synthesis conditions ensure complete control of the morphological, electronic, and physicochemical properties of resulting GY sheets as well as the distribution of dopants deposited on GY surfaces. The uniform and quantitative inclusion of non-metallic (B, Cl, N, O, or P) and metallic (Fe, Co, or Ni) elements into graphyne derivatives were theoretically and experimentally studied, which improved their electronic and magnetic properties as row systems or in heterojunction. The effect of heteroatoms associated with metallic impurities on the magnetic properties of GYs was investigated. Finally, the flexibility of doped GYs’ electronic and magnetic features recommends them for new electronic and optoelectronic applications.

## 1. Introduction

Among all chemical elements, carbon exhibits the greatest flexibility of its first coordination sphere, which is usually presented in textbooks as sp, sp^2^, and sp^3^ hybridizations. This plasticity leads to three available types of bonds (single-, double-, and triple-bonded C atoms) that may occur in diverse, practically unlimited, connectivities. Altogether, this markedly influences the allotropy of carbon, which is the richest among all chemical elements. Familiar allotropic forms of carbon include graphite, rhombohedral graphite, diamond, lonsdaleite, amorphous carbon (soot with variable sp^2^/sp^3^ carbon atom contents), and a huge variety of human-made high-specific surface area carbons, carbon aerogels [1,2], carbon foams [3,4], glassy carbon [5,6], polyynes [7], diverse fullerenes from C_12_ to as large as C_960_ [8,9], a multitude of single wall nanotubes [10,11], nano-onions [12], and more. Last but not least, they include graphene, which has unique physical properties [12]. Other exotic forms such as ultra-high pressure BC8 [13,14], Po-32 [15], and ferromagnetic carbon [16] have previously been theorized [17] (C18 was even reported [18]), but some are still disputed. Nevertheless, this structural diversity and versatility of chemical bonding brings an enormous pool of physicochemical properties, reactivities, and so on; crystal structures of over 500 periodic allotropes, known and hypothesized, have been collected in a unique Sacada database (https://www.sacada.info/) [19]. Despite the long-lasting research of carbon-based materials, some fundamental issues related to the shape of the phase diagram and mutual stability of polymorphs, or even their existence, remain unresolved to this day [20,21,22]; e.g., it has been recently claimed that lonsdaleite is not a genuine allotropic form but a twin of cubic crystals, which raised controversy [23,24]. One illustration of the intensity of the research field of carbon materials can be provided by an inspection of the Web of Science database; this resource lists approximately 114,000 papers using the keyword ‘diamond’, approximately 147,000 papers discussing ‘graphite’, and approximately 240,000 papers featuring ‘graphene’. It can be safely estimated that—even given some overlap—well over half a million of over 90 million indexed scientific works (1900–2021) have been devoted to carbon allotropes. The discovery of graphene triggered the production of diverse 2D carbonaceous materials including graphone, graphane, and graphene oxide [25,26,27]. Graphynes (GYs) are layered two-dimensional structures built from sp- and sp^2^-hybridized carbon atoms (Figure 1A).

Wide tunability in structural, mechanical, physical, and chemical properties make GYs fascinating candidates for use in energy storage, solar cells, electronic and spintronic devices, UV light detectors, as well as adsorbents in the separation of gases [28]. A GY can be a useful catalyst in water purification [28]. GYs have shown improved electronic properties and charge carriers in the optics and electronics industries. Graphone and graphane are hydrogenated forms of graphene—with fundamental band gaps of 2.45 and 5.4 eV, respectively, and interesting magnetic properties—that have shown potential for nanoelectronics and spintronics [29,30,31]. Due to structural similarities with graphene, these materials are considered to be excellent candidates for carbonaceous electronic devices, and they surely will be the subject of advanced studies for multiple and versatile applications. Scientists have been theoretically studying graphynes since the 1980s, which initially attracted attention after the discovery of fullerenes [32]. Though the structure of GY was first proposed in 1987 by Baughman et al. [33], the demanding synthesis of GYs hindered their dynamic development for more than 20 years [34]. Among all the GYs, the rising-star γ-graphdiyne (γ-GDY), seen in Figure 1B, was the first GDY member experimentally synthesized and reported by Li et al. in 2010 [35]. Because of the promising physical, optical, and mechanical features of GYs, a tremendous amount of research effort has been dedicated by theoretical, applied, and synthetic chemists. It is believed that GYs might pose competition for more common sp^2^-hybridized carbon systems, particularly graphene, and meet the increasing demand for an alternative candidate to carbonaceous materials. Recent years have brought a sharp increase in research interest on the synthesis and theoretical prediction of GYs’ properties in different dimensions, e.g., one-dimensional nanowires (such as nanotube arrays and ordered stripe arrays), and two-dimensional nanowalls (2D) and nanosheets (2D), and 3D frameworks [36,37,38]. More and more works are concerning pristine GYs’ structures enriched with diverse heteroatoms (B, F, N, O, etc.) [37]. Due to the broad spectrum of intense scientific activities related to the development of new forms of GYs, this work emphasizes recent developments in this field, especially dealing with newly obtained heteroatom-doped structures and investigating the synergic effects of heteroatoms, metal oxides, and metal ions on the electronic and optoelectronic properties of doped GYs. In this work, we also focus our attention on theoretical and experimental research into the magnetism of pristine and doped GYs.

## 2. From Atomic Structure Suggestion to Experimental Appearance

In 1987, Baughman et al. [33] proposed a novel two-dimensional allotrope of carbon assembled by aromatic centers (sp^2^) linked to each other by acetylenic bridges (sp); these were named graphynes (GYs). In the beginning, GYs (a general name given to this type of material) were designated according to the number of carbons included in the various rings forming a given network (shaped pores in the structure). For example, A, B, and C networks were called 18,18,18-graphyne (Figure 2A), 12,12,12-graphyne (Figure 2B), and 6,6,6-graphyne (Figure 2C), respectively. The latter theoretically present the lowest energy of carbon phases consisting of flat molecular sheets obtained by inserting acetylene linkers between the aromatic rings into the pristine honeycomb graphene structure. As energetically favorable, the 6,6,6-graphyne has a special place in the GY family, and the term “graphyne” without specifying the number of carbons is reserved for this phase [39].

In 1997, Haley et al. proposed new forms of GYs called graphdiyne (GDY), a nanostructure formed from two adjacent acetylenic linkers between aromatic carbon atoms in contrast to graphyne, which is made of carbons connected by one triple bond [40]. Therefore, to investigate new structures, different GY networks designations were developed and used in the literature [39,41]. Figure 2 shows several types of GY structures named in accordance with the widespread terminology including α-graphyne (Figure 2A), β-graphyne (Figure 2B), γ-graphyne (Figure 2C), α-graphdiyne (Figure 2D), β-graphdiyne (Figure 2E), and γ-graphdiyne (Figure 2F). The last nomenclature method named γ-graphynes with various numbers of acetylenic linkers in a structure, such as graph-n-yne (*n* = 1, 2, 3…, where *n* is the number of triple bonds). The mentioned nomenclature methods properly operate when naming the most conventional GYs and GDYs. However, the graphyne naming system causes considerable problems and should be standardized, especially for structures containing heteroatoms. In this paper, GYs, GDYs, and their derivatives are all briefly designated as graphyne- or graphdiyne-like structures with the abbreviation “GYs” and “GDYs”, respectively.

In order to prepare extended structures of GYs, chemists developed versatile synthetic methods grouped into two subsections: top-down and bottom-up [42]. The top-down methods have the potential to obtain two dimensional structures from their bulk precursors through processes such as mechanochemical synthesis and vapor-liquid-solid (VLS) growth. The bottom-up formulations result in the formation of thin films or a few layers of GYs through coupling reactions of desired substrates in carefully chosen and strictly controlled conditions. “On-surface” coupling synthesis conducted in an ultra-high vacuum (UHVS), chemical vapor deposition (CVD), thermal treatments, and wet-synthetic approaches are practical subdivisions of the bottom-up method.

By summarizing the most representative and effective synthetic methods, we hope they can create an idea and solid foundation for preparing GYs with given applications in various research fields. The wet-synthetic strategy is the most successful and high-demand method for the preparation of a wide range of applicable GYs, in both pristine and doped natures. In order to facilitate understanding and comparison, the wet-synthetic methods are classified into two categories based on the different phases where the coupling reaction of the reactants occurs amid the synthesis process: one-phase (homogenous reaction) and two-phase (interfacial reaction). Two-phase methods have three sub-divisions: liquid/liquid, solid/liquid, and gas/liquid.

We intend to summarize the developments in preparation methods resulting in different GY morphologies (nanowires, nanotubes, nanowalls, nanoribbons, nanosheets, etc.), as well as different chemical compositions. The broad-spectrum synthesis protocols employed in cross-coupling or homocoupling reactions of graphynes’ subunits and derivatives were named after their inventors as Glaser [43], Eglinton [44], Hay [45], Negishi [46], Hiyama [47], and Sonogashira [48] reactions. All these methods, briefly presented in Table 1, were proven to be applicable to the preparation of valuable and scientifically important heteroatom-doped GYs.

Theoretically proposed graphynes (GYs) induced tremendous effort to find applicable routes, resulting in the fabrication of these materials [39]. The controlled oligo-trimerization of cyclo-carbons such as cyclo-C18 (cycle made of nine acetylenic groups) and cyclo-C12 (cycle made of six acetylenic groups) and the oxidative polymerization of monomeric acetylenic precursors or synthetic macrocyclic compounds under Glaser–Hay coupling conditions are potential methods for the synthesis of γ-GDY, γ-GY, β-GDY, and other derivatives [49,50,51,52]. As an alternative strategy to investigate the potential properties of GYs, Haley et al. proposed a method for the synthetic preparation of γ-GY and γ-GDY substructures [40,53]. These subunits can be next used as the first block in the construction of extended GY structures. This breakthrough in synthetic approaches to alkynyl carbon materials revealed possibilities for the synthesis of low-dimensional carbonaceous nanomaterials involving acetylenic scaffolds [40,54,55,56,57]. Since the first report of γ-GDY preparation in 2010, exciting progress has been made in the experimental preparation of GYs. In the next sections, we describe different, well-developed synthetic protocols leading to graphyne-like structures composed of C, as well as structures chemically modified by heteroatoms (e.g., N, F, Cl, H, S, and B).

### 2.1. Mechanochemical Synthesis

A mechanochemical synthesis route is used in one of the most effective methods. Its strength lies in its simplicity, rapidity, and repeatability. As a result, solid extended GYs or their subunits with reproducible features may be produced. Under mechanical impact and at an elevated local temperature resulting from particle collisions, selected bonds of substrates are broken and new compounds may form in the solid-state, thus overcoming the problems associated with solution-based chemistry processes.

The reaction of hexachlorobenzene (HCB; known as a persistent organic pollutant) and calcium carbide (CaC_2_; known as an efficient and safe co-milling reagent) in a planetary ball mill at room temperature within 20 min of milling at a mass ratio of CaC_2_/HCB = 0.9 and a rotation speed of 300 rpm, as proposed by Li et al. [58] in 2017, caught the attention of materials chemists in the preparation of GYs. After this development, a mature mechanochemical approach was applied by Li et al. in 2017 for the one-step high-yield synthesis of GY monomers [59]. One year later, Cui et al. synthesized hydrogen-substituted graphyne (H-GY) and γ-graphyne (GY) via the ball-milling-driven mechanochemical cross-coupling of 1,3,5-tribromobenzene (PhBr_3_), hexabromobenzene (PhBr_6_), and CaC_2_ as precursors under a vacuum. Finally, the impurities were removed with diluted nitric acid and benzene (Figure 3) [60,61].

### 2.2. The Vapor–Liquid–Solid Growth

Synthesis reactions carried out with vapor-liquid-solid (VLS) growth allow one to control the production of different types of nanomaterials. In general, GY sheets are harvested on carefully prepared surfaces of monocrystalline silicon coated by metallic nanoparticles (Au, Fe, Zn, and Ni). The prepared substrate-catalyst support provides favorable physical features (surface energy, stability, and crystal structure) for the effective growth of nanostructural materials. The VLS method was applied by Li et al. for graphdiyne (GDY) film synthesis [62]. In their approach, powdered and vaporized GDY was deposited on the surface of ZnO nanorods grown on a silicon wafer (Figure 4a). Liquid nanodroplets of melted zinc (~419 °C) formed on one of the ends of the ZnO nanorods (ZnO NRs) (Figure 4b) served as energetically favorable adsorption sites of incoming vapors of GDY molecules. Due to the mixing of vapors with melted zinc, a solution of graphdiyne and Zn was formed (Figure 4c). The continuous influx of fresh portions of GDY molecules resulted in the formation of a supersaturated solution (GDY-Zn), as well as the fusion of drops and an increase in their size of droplets, thereby facilitating lateral growth due to the small edge energy of 2D materials.

### 2.3. Thermal Treatment

The heating of hexaethynylbenzene (HEB), N-rich precursors (2,4,6-triethynyl-1,3,5-triazine, TET), and pentaethynylpyridine (PEP) was applied by Zuo et al. to force a homocoupling reaction, resulting in GDY nanostructures with different nitrogen percentages and morphologies (nanoribbon, nanochain, and 3D-networks) [63,64]. Notably, this reaction could be carried out without using any metal catalyst. The powder of N-rich precursors were slowly delivered to the preheated conical flask (120 °C), leading to an explosive reaction whereby black GDY was obtained (Figure 5a,b). A gradual heating process (10 °C/min) to 120 °C in nitrogen converted the light-yellow HEB into black nanoribbon-like morphologies without volume change (Figure 5c,i). On the other hand, the implementation of this treatment in an air atmosphere resulted in GDY nanochains uniformly grown on the 3D network with a remarkable volume increase of 6 times (Figure 5c,ii). This means that the oxygen accelerated the dehydrogenation for the coupling reaction. However, the addition of HEB into a preheated air environment (120 °C) rapidly caused a more violent reaction, and an ultrafine nanochain with a 48-fold volume increase was obtained (Figure 5c,iii).

The doping of carbonaceous materials with N atoms can be realized via the three following routes: chemical vapor deposition (CVD), the pyrolysis (annealing) of N-containing precursors, and heating with N-rich chemicals. For instance, N-doped GDY, B-doped GDY, F-doped GDY, and S-doped GDY have been prepared using an annealing strategy [65,66,67,68,69,70]. In such a case, GDY-based nanomaterials have been thermally treated with relevant chemicals, such as ammonia (N), B_2_O_3_ (B), NH_4_F (F), and thiourea (S), at a chosen temperature [70]. The considered methods, however, suffer from drawbacks, such as the randomness of the doping sites and uncontrolled dopant’s percentage. Therefore, there is still a need to develop a synthetic strategy that provides 2D carbon materials with a homogeneous distribution of atoms of the desired type and at specific desired locations.

### 2.4. “On-Surface” Synthesis under an Ultra-High Vacuum

The so-called “on-surface synthesis” approach is carried out under an ultra-high vacuum (UHV) and presents considerable potential in building new types of nanomaterials. In this method, the starting building blocks are deposited onto the surface of a metallic substrate (Ag and Au) [71,72,73,74,75,76,77,78,79,80,81,82,83,84,85]. Then, the coupling of precursors occurs, resulting in single-atom-thick 1D and 2D materials. In contrast to conventional “wet” reactions, it helps here to eliminate possible undesired influences from surroundings. The chemical character of the organic precursors and the nature of the applied substrate are crucial factors that determine the final properties of GDY nanostructures [41]. Features such as the dimensionality of the organic precursors, the reactivity of their functionalities, the geometry of the surface, and interactions occurring between the organic building blocks and substrate have significant effects on the reaction and molecular surface patterns. The reactivity and mass of the used molecules have significant impacts on the success of synthesis. High reactivity or weight may prevent them from sublimating on the surface of the substrate. On the other hand, molecules that are too small will escape from the reaction chamber. The metallic substrates play a double role as templates and catalysts of the coupling reaction. In the following sections, we discuss the most effective method for the synthesis of graphyne analog (sub-structures or infinite nanostructures) based on “on-surface” coupling reactions.

The production of acetylenic frameworks at interfaces involving the formation of self-assembled monolayers (SAMs) followed by a cross-linking step to form linked monolayers is a direct way to create carbonaceous materials such as carbyne, graphyne, and graphdiyne [71,72]. “On-surface” homocoupling reactions requiring the detachments of halogens or hydrogen from precursors functionalized with alkynyl groups have been reported as effective fabrication methods of low-dimensional carbon-based nanostructures. An overview of the production of 1D carbonaceous nanomaterials via “on-surface” approaches was previously described in the literature [73].

Rubben et al. reported a surface-assisted dehydrogenative homocoupling reaction of terminal alkynes (Csp–H), such as triethynylbenzene (TEB) and (1,3,5-tris-(4-ethynylphenyl)benzene (Ext-TEB), that was conducted on a Ag(111) surface, wherein the hydrogen was the only by-product of the reaction. The authors stated that it was an ideal method for the synthesis of individual chemicals or polymeric structures containing a conjugated backbone (after annealing at 400 K) [74]. In 2015, Wu et al. investigated the reaction of 2,5-diethynyl-1,4-bis (phenylethynyl)-benzene (DEBPB) taking place on the surface of silver with different facets including (111), (110), and (100). The reaction was carried out with the aid of scanning tunneling microscopy (STM). The Glaser synthesis conducted on Ag(111) was dominant and yielded one-dimensional, covalently bonded wires. On the contrary, reactions conducted on Ag(110) and Ag(100) surfaces resulted in one-dimensional organometallic frameworks built on terminal alkynes and metal atoms. (Figure 6) [75].

Klappenberger et al. employed a Ag(877) support to obtain one-dimensional conjugated molecular threads as components of extended GYs with lengths reaching 30 nm [76]. Thermal dehydrogenative reactions carried out on a flat Ag(111) plate were found to be associated with several undesirable side reactions that resulted in the formation of branched, irregular nanostructures. Liu et al. induced a dehydrogenative reaction of 2,5-diethynyl-1,4-bis(4-bromophenylethynyl)benzene and noticed that bromine adatoms affected the activation of C–H groups in terminal alkynes occurring at 298 K on a Ag(111) surface [77]. The STM studies disclosed the formation of organometallic species followed by their partial conversion to covalently bonded nanostructures after annealing at 420 K.

In 2018, Xu et al. applied a dehalogenative homocoupling reaction to tribromoethylbenzene (TBP), 1,3-bis(tribromomethyl)benzene (bTBP), and 1,3,5-tris(tribromomethyl)benzene (tTBP), and they converted tribromomethyl functional groups (Csp^3^) to form C–C triple bonds (Csp) as structural motifs of dimeric structures, such as wires or 2D networks of GYs grown on a Au(111) surface [78].

In 2020, Xu et al. created on-surface graphyne nanowires through dehalogenative homocoupling reactions via the stepwise activation of two different types of C–Br bonds (involving Csp^3^–Br and Csp^2^–Br) in a 1-bromo-4-(tribromomethyl)benzene (BTBMB) compound on both Au(111) and Ag(110) surfaces [79]. Sun et al. also reported the successful formation of dimer structures with acetylenic linkers (wires and networks) via the on-surface C–Br activation of alkenyl carbon atoms [80]. Two-dimensional networks with acetylenic linkages were obtained after the homocoupling reaction of 1,3,5-tris(bromoethynyl)benzene (tBEP). In the first step, the precursor was deposited on a Au(111) surface at room temperature and was slightly heated up to 320 K. As a result, organometallic networks were obtained. The further increase in temperature up to ∼450 K led to the release of gold atoms and the formation of the final product. Figure 7 shows STM studies of this reaction. Considering the published results, the Ag(111) surface seems to be the most effective substrate for Glaser coupling because it causes a smaller number of side reactions [81] than Au(111) plates, for which the cyclization of the terminal alkyne to the benzene ring is common [82,83,84]. When used as a substrate, a Cu(111) surface presented low activity towards “on-surface” Glaser coupling. This stands in contrast to wet coupling reactions for which Cu ions show very high catalytic activity, and Cu is regarded as the most effective catalyst for such reactions [85].

### 2.5. “On-Surface” Synthesis by Chemical Vapor Deposition

Another synthesis strategy based on the covalent coupling of organic monomers occurring at the metal surface, so-called “on-surface” synthesis, is chemical vapor deposition (CVD).

Furthermore, the CVD method is recognized as one of the most promising routes for the creation of novel 2D materials. This approach relies on the transfer of vapors of monomers to reaction chambers and their embedding and coupling on the surface of a preheated metallic substrate. However, this method has severe limitations. As the reaction is conducted on a metal substrate without any additional catalysts, the reaction stops when the surface is fully covered with GY monolayer films (Figure 8) [86]. It has been statistically shown that silver seems to be the most efficient substrate for carrying out such reactions. In contrast to other investigated metallic substrates (such as Au and Cu), silver was found to ensure the lowest proportion of side reactions [86].

### 2.6. Wet Chemical Synthesis

Since the first successful fabrication of γ-GDY through a liquid/solid interfacial reaction on Cu foil as a catalytic substrate for an acceleration coupling reaction in an organic solvent, directed efforts have been carried out to prepare graphyne derivatives. Homo-coupling and cross-coupling reactions of hexaethynylbenzene (HEB) as an efficient precursor have been realized as operational pathways for the preparation of graphyne analogs (fragments, oligomers, or infinite structures) and have recently attracted immense attention. These methodologies employ metal salts (copper salts: Cu(I), Cu(II), or Pd(II)) in a homogenous conditions or on a surface template, e.g., Cu foil or other arbitrary surfaces such as graphene, Au, and 3D foam (which may have catalytic properties) to assist in a heterogeneous reaction [41].

#### 2.6.1. Developments of Coupling Reactions (All Reactants in Solution Phase)

The coupling reaction for the synthesis of graphyne analogs proposed by Moroni et al. combined dibromoaryls (I) and 1,4-diethynylaryls (II) in the presence of PdCl_2_, Cu(OAc)_2_, and triphenylphosphine (PPh)_3_ in a triethylamine/THF mixture (Figure 9A), where R1, R2, R3, and R4 could be the same or different (H, NO_2_, alkyl ether, alkyl thioether, or alkyl ester). Homopolymers or copolymers with phenyl, thienyl, anthryl, or stilbene groups as aryl units were synthesized [87]. Despite this progress in synthetic methods, extended structures of GYs are still unachievable.

The combined Negishi and Sonogashira cross-coupling reactions for the formulations of various kinds of substituted hexaethynylbenzenes from chloroiodobenzenes put researchers on a fast track towards the fabrication of GYs [88,89]. In 2007, Jiang et al. obtained poly(aryleneethynylene) networks with highly developed porous structures. To do so, they applied a Pd-supported Sonogashira-Hagihara reaction, which had previously been employed to synthesize different polymeric compounds such as polymers and ligands for coordination-polymer synthesis, wires, and shape-persistent macrocycles [90]. In 2010, Dowson et al. showed that porous properties (BET surface area and pore volume) are strictly controlled by the kind of solvent used as the environment of the reaction [91]. Toluene, tetrahydrofuran (THF), N-dimethylformamide (DMF), and 1,4-dioxane were tested for these reactions. Authors showed that DMF is the most proper solvent, as its received nanostructures are characterized by the highest BET surface areas (up to 1260 m^2^/g) [91,92]. In homogenous coupling reactions, materials chemists prepared oligomers and macromolecules, but infinite structures are still elusive.

After 2010, remarkable progress in GY preparation was achieved. The synthesis of poly(aryleneethynylene)s (PAEs) using Pd and Mo/W was thoroughly investigated by Bunz in 2010 [93]. Wu et al. utilized commercially available tris(*t*-butoxy)(2,2-dimethylpropylidyne)-tungsten(VI) as the catalyst in the synthesis of hydrogen-substituted graphyne (H-GY), as shown in Figure 9B [94]. Ding et al. prepared γ-graphyne in a homogenous ultrasound-driven reaction of hexabromobenzene (PhBr_6_) and calcium carbide (CaC_2_) in an inert atmosphere without a metal catalyst [95]. Wen et al. prepared new N-doped graphyne analogs (Figure 9C) in the reaction of nucleophilic substitution (SNAr) of cyanuric chloride and para-dilithium aromatic reagents. The process was carried out under mild conditions in a diglyme or bis(2-methoxyethyl) ether (solvents with high boiling points) solution. That designed reaction allowed them to obtain N-GYs on a gram scale [96]. The development of GDY synthesis was a breakthrough in the preparation of different morphologies such as films, nanowires, nanotube arrays, nanoribbons, nanosheets, and nanowalls of GYs with versatile properties, as well as reductions in the dimensionality [97].

**Figure 9 nanomaterials-11-02268-f009:**
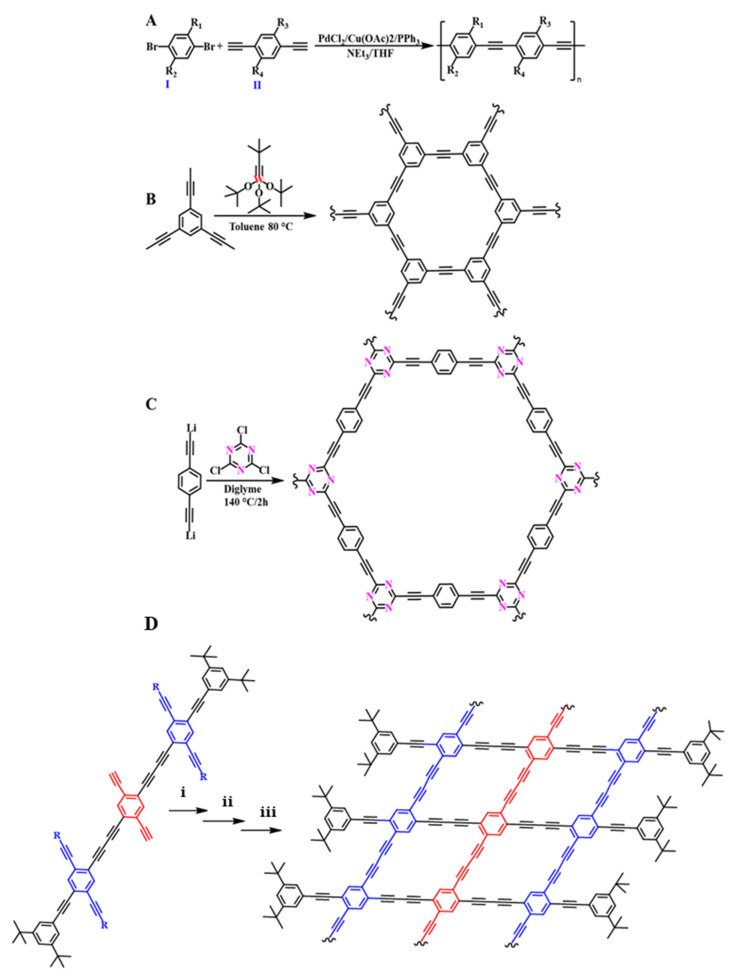
Homogenous coupling reactions in the preparation of graphyne derivatives (**A**). Synthetic scheme of the graphyne (**B**) and N-doped graphyne-like nanostructures (**C**). Redrawn [87,94]. (**D**) Intermolecular Glaser-Hay cross-coupling reactions of red fragments for GDY nanoribbons by copper(I) chloride, TMEDA, and acetone/tetrahydrofuran at RT (i); tetra-n-butylammonium fluoride and THF at RT (ii); intramolecular coupling reaction of blue fragments by Cu(OAc)_2_ and pyridine, H_2_O, and CH_2_Cl_2_ at RT (iii); R = triisopropylsilyl; TMEDA: *N*,*N*,*N*′,*N*′-tetramethylethane-1,2-diamine; RT: room temperature. Redrawn [63,64].

Graphdiyne nanoribbons (GDYNRs) comprise a class of 1D GDY materials that stresses well-defined edges and nanometer size [97]. There have already been numerous theoretical efforts regarding GDYNRs seeking connections between their structures and properties. The results of this research are discussed in Section 3.2.3. A bottom-up chemical formulation could provide structurally uniform and well-defined nanostructures of GDYNRs. It is, however, necessary to perform the selective stepwise coupling of ethynyl groups during the synthesis procedure. Zhou et al. proposed a two-step method of intermolecular polymerization followed by the intramolecular cross-coupling of acetylenic moieties, as seen in Figure 9D (red fragment). First, the polymerization of the ethynyl units in the central part of the monomer ensures one-dimensional growth (Figure 9D, red fragment). Secondly, the intramolecular reaction of ethynyl groups on the established facing side chains and the bulky groups (such as the 3,5-di-tert-butylbenzyl group) on the outer side occurs. The latter works to sterically hinder intermolecular coupling. This strategy was applied to build GDYNRs nanostructures (Figure 9D, blue fragment) made of rhomboid units with benzene as junctions and butadiyne as linkers for the first time. The structures showed a well-defined width of ~4 nm and a length of hundreds of nanometers [97].

#### 2.6.2. Two-Phase Methods (Interfacial Synthesis Utilizing Two Immiscible Liquids)

Atomic, ionic, or molecular compounds may be successfully applied as starting materials to the direct, bottom–up synthesis of ultrathin GDY nanostructures with in-plane periodicity. In 2017, Sakamoto et al. strived to create graphdiyne at the interface between two immiscible fluids (Figure 10A,B) [98].

The upper aqueous phase held copper (II) acetate and pyridine, which catalyzed ethynyl homocoupling (Eglinton coupling). The lower dichloromethane phase contained the HEB monomer. The continuous catalytic reaction for 24 h under an inert atmosphere at room temperature resulted in the development of a layered GDY (thickness: 24 nm; domain size: >25 µm). In 2019, Song et al. described the liquid/liquid interfacial formulation as a comprehensive way to obtain GYs via a reaction between terminal ethynyl groups and an aryl halide. The reactions catalyzed by PdCl_2_(PPh_3_)_2_ and CuI resulted in various forms of GY nanostructures, including hydrogen-substituted graphyne (H-GY), methyl-substituted graphyne (Me-GY), and fluorinated graphyne (F-GY) [99]. H-GY is a framework consisting of duplicated sections of benzene rings joined by ethynyl linkers at all meta sites. Likewise, the repeating units of Me-GY (or F-GY) are 1,3,5-trimethylbenzene (or 1,3,5-trifluorobenzene) rings attached to benzene rings within acetylene bridges. Two-dimensional N-graphdiyne sheets were recently prepared via reactions conducted at the interfaces (Figure 10D,E) [100]. Nitrogen heterocycles (triazine and pyrazine) bearing terminal ethynyl groups were polymerized through Glaser coupling reactions at interfaces. This procedure was expanded to the synthesis of S-doped graphdiyne (TTF-GDY) structures comprising tetrathiafulvalene fractions (Figure 10C) and has this potential to be applied as a robust route for the synthesis of a wide range of heteroatom-rich graphyne-like structures in the future [101].

**Figure 10 nanomaterials-11-02268-f010:**
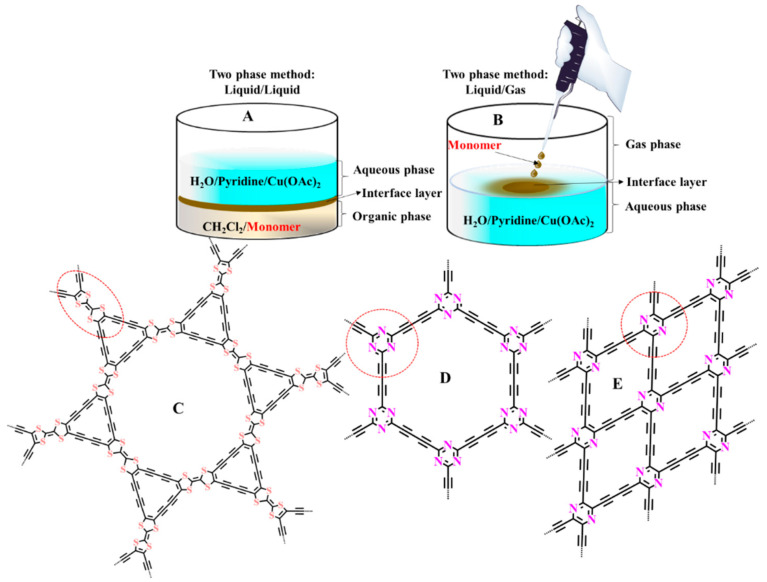
Homocoupling reaction in interlayer between two phases: (**A**) liquid/liquid and gas/liquid (**B**) in the preparation of sulfur-rich graphdiyne (**C**) and N-graphdiyne (**D**,**E**). Redrawn [100,101].

#### 2.6.3. Developments of Heterogeneous Coupling Reactions at Liquid/Solid Interfaces on Diverse Substrates

In liquid/solid interface reactions, substrates such as copper foil, plate, foam, and walls have been applied to bring reactants together and speed up the reaction procedure [102,103]. The first γ-GDY was prepared through the heterogeneous homo-coupling reaction of hexaethynylbenzene on copper foil, playing double roles of catalyst and substrate (Figure 11A) [28]. It was reported that both the oligomer evaporation process and the kinetics of the coupling reaction were strictly controlled by temperature [104,105]. Furthermore, the factors determining the structural properties of the obtained nanostructures were found to be catalyst distribution and monomer concentration. The formation of γ-GDY nanowalls via the Glaser-Hay reaction was successfully carried out in the presence of *N*,*N*,*N*’,*N*’-tetramethylethylenediamine (TMEDA) due to its ease in complexing copper ions. The copper envelope catalysis strategy was employed for the synthesis of γ-GDY nanowalls on various substrates, including one-dimensional Si nanowires; two-dimensional Au, Ni, and W foils; quartz; 3D stainless steel mesh; and 3D graphene foam (GY), as seen in Figure 11B [106]. For this purpose, these were the chosen substrates used in this method, and the target substrates were wrapped in a Cu-based envelope.

Well-defined films of triazine-based graphdiyne (TA-GDY), aminated-graphdiyne (NH_2_-GDY), β-GDY, H-GY, Cl-GDY, and F-GDY were prepared through Glaser homocoupling reactions on Cu foil (Figure 11C) [107,108,109,110,111,112]. 2,4,6-triethynyl-1,3,5-triazine, 2,4,6-triethynylaniline, tetraethynylethene (TEE), 1,3,5-triethynylbenzene, 1,3,5-trichloro-2,4,6-triethynylbenzene, and 1,3,5-triethynyl-2,4,6-trifluorobenzene were the respective starting compounds (prepared from the deprotection of trimethylsilyl group by tetra-butyl ammonium fluoride (TBAF) from corresponding silylated substrates) and were applied in homocoupling reactions to prepare extended GYs on Cu foil. The resulting films were peeled off from the Cu foil by a FeCl_3_-saturated solution and then rinsed with H_2_O, acetone, DMF, and ethanol. Wang et al. prepared boron-doped graphdiyne (B-GDY) through the aforementioned synthetic procedure, which was recognized to be an effective technique to prepare 2D carbonaceous nanostructures with strictly controlled and well-organized chemical structures (Figure 11D) [113]. In contrast to the copper foil, the copper nanowires (CuNWs) worked as templates and delivered more reactive sites for developing γ-GDY structures [114,115]. As a result, high-quality nanostructures with well-developed surface areas were obtained. For instance, the thin films of graphdiyne (average thickness: approximately 1.9 nm) were obtained on CuNWs (100 nm in diameter) [114]. In the end, polymeric films were isolated by washing a crude product in a mixture of hydrochloric acid and FeCl_3_.

## 3. Electronic Properties

### 3.1. Dirac Cone

A Dirac cone is a distinctive feature in an electronic arrangement in which the energy levels of the valence and conduction bands meet at one specific point in the first Brillouin zone (named Dirac points), hence setting the Fermi level; the band structure in its vicinity resembles a double cone with linear dispersion [116]. The presence of a Dirac cone renders a given material “the zero-gap semiconductor” rather than metal and results in several unusual features such as ballistic electronic transport and enormous thermal conductivity. The occurrence of Dirac cones in graphene, predicted by Wallace in 1947 and experimentally demonstrated by Novoselov et al. in 2005, has sparked unceasing research in recent years [117,118]. Some efforts have been directed towards exploring the possibility of the occurrence of the Dirac cones in GYs. The presence of Dirac cones in *α*-graphyne, *β*-graphyne, and γ-graphyne-n with hexagonal symmetry structures was demonstrated. Recently, Vines et al. discovered that 6,6,12-GY, with rectangular symmetry, has two self-doped non-equivalent and distorted Dirac cones [119,120]. These results shed new light on the electronic properties of GY-like materials and suggest that rigorous hexagonal symmetry is not a feature that determines the appearance of cones in these materials.

### 3.2. Electronic Band Structure

The electronic band structures of diverse GYs and GDYs have been investigated using theoretical methods, particularly density functional theory (DFT). Local density approximation (LDA) and the generalized gradient approximation (GGA) have been widely applied to study the structural, mechanical, electronic, and magnetic properties of GYs. Nevertheless, the underestimation of the band gap levels still is one of the critical problems of LDA and GGA. Hybrid Heyd-Scuseria-Ernzerhof-type functionals (HSE) have improved total energy evaluation by admixing the nonlocal Hartree-Fock exchange. They also lead to more realistic band gaps than LDA or GGA functionals. However, those conducted calculations generate significantly higher computational costs. To describe the van der Waals force in vdW-optPBE layered compounds, some extent of correction leads to improved results. Next, we review theoretical studies on GYs and compare diverse calculated parameters with the aforementioned theoretical approaches.

#### 3.2.1. The Electronic Band Structure of GYs

The optimized geometry and electronic structures of diverse GY materials (graphyne, graphdiyne, graphyne-3, and graphyne-4) were computed by applying the full-potential linear combination of atomic orbitals (LCAO) approach by Narita et al. in 1998 [121]. The unit cell of all considered GY derivatives was similar to graphyne and is shown as a parallelogram in Figure 12. This unit cell was found to contain 12 carbons, the a and b lattice vectors were found to be equal (a = b), and the angle between them was found to be γ = 120°. The Brillouin zone of the investigated material was an equilateral hexagon. As a result of geometry optimization, all bond angles are either 120° or 180° in graphyne-n. The lattice parameters of graphyne-n structures (*n* = 1, 2, 3, and 4) are 6.86, 9.44, 12.02, and 14.6 Å, respectively, and binding energies are 7.95, 7.78, 7.70, and 7.66 eV, respectively. It turned out that the reported hexagon presents a bond length almost equal to those found for graphite. Moreover, it is a bit longer than the bond that extends outside a hexagon. The bridges between hexagons are not formed by cumulenic linkers =C=C=; rather, they are formed by ethynyl ones (–C≡C–). The presence of conjugated multiple bonds is a typical feature of graphyne and its derivatives.

The indirect band gap is the separation between the conduction band minimum (CB_min_) and the valence band maximum (VB_max_) within an electronic band structure, whereas the direct band gap denotes the smallest of gaps at one particular point in the Brillouin zone [122]. Chen et al. studied the band structures of optimized α-, β- and γ-GY structures (Figure 12A) [123]. They showed that all investigated materials were direct semiconductors. In the case of α-graphyne, the conduction band minimum meets the valence band maximum at K-point in the Brillouin zone (Figure 12Ba) and symmetric Dirac cones are formed. In the case of β- and γ-GY, the energy bands coincide at M-points in the Brillouin zone (Figure 12Bb,c) and show quasi-Dirac cone structures, albeit with a slightly open band gap. The values of band gaps were found to be equal to 0 eV (α-GY), 0.028 eV (β-GY), and 0.447 eV (γ-GY) when applying the generalized gradient approximation of the Perdew–Burke–Ernzerhof (GGA-PBE) method (Figure 12Ba–c). Moreover, *γ*-GDY is a direct semiconductor with a Dirac point at the zone center (Γ point of the first Brillouin zone).

**Figure 12 nanomaterials-11-02268-f012:**
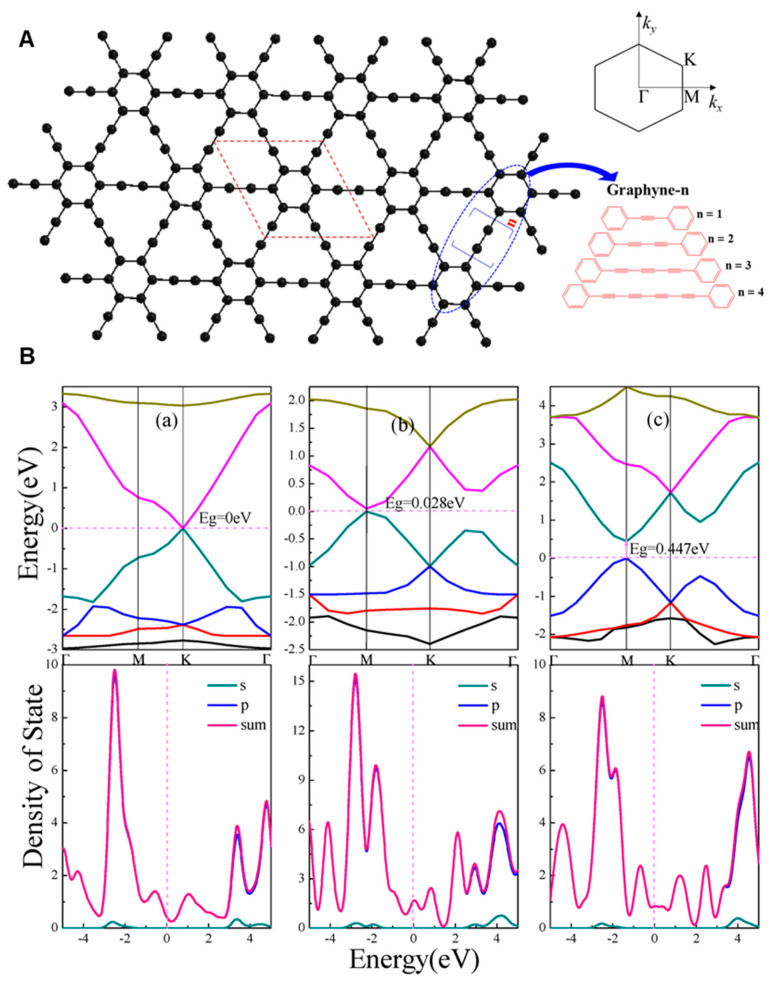
Illustration of graphyne-n structures. The red parallelogram shows the unit cell of GYs. The Brillouin zone is shown in the upper-right, and various acetylenic linkers are shown in the bottom-right (**A**); (**B**) band gaps (Eg) and density of states (DOS) of α-graphyne (**a**), β-graphyne (**b**), and γ-graphyne (**c**). s and p describe the partial densities of states of the s and p orbitals in carbon atoms, respectively. The sum of both elements gives a value of the total density of states. Reproduced with permission [123]. Copyright 2018, MDPI.

The estimated band gaps of graphyne-like families (monolayer, bilayer, multilayer, nanotubes, and nanoribbons) with and without strain by different functionals are compiled in Table 2. A wide range of band gap levels were found for *γ*-GY [121,122,123,124,125,126,127,128,129] and *γ*-GDY [42,106,121,125,126,127,130,131,132,133,134] (0.447–2.23 eV for *γ*-GY and 0.44–1.21 eV for *γ*-GDY), which is strictly related to the applied calculation functionals, as reviewed in Table 2.

The manipulation of the band structure of GYs has been attempted through strategies including strain tuning, structural engineering (the fabrication of GYs with different dimensions), doping, and the application of external electric fields. The obtained results suggest that GYs are valuable materials for nanoelectronic and optoelectronic devices or sensors. For example, strained graphyne-n was obtained by applying different types of strain, including homogeneous biaxial (H-strain in both the x and y directions) and uniaxial strains (x-direction A-strain and y-direction Z-strain), as was reported by Li et al. in 2013 (Figure 13a) [127]. For the H-strain case, hexagonal symmetry was preserved after the strain (Figure 13b). The A-strain deformation was found to be in the direction of the propagation of the ethynyl linkers, while the Z-strain described deformation perpendicular to the acetylenic linkers with bent hexagonal symmetry (Figure 13c,d). Unlike graphene, which exhibits a band gap insensitivity to applied strain, GYs have shown band gap modulations under different straining approaches. Comprehensive studies have proven that homogeneous tensile stress expands the band gap of GYs, whereas uniaxial tensile and compressive strains lead to band gap decreases (Figure 13e). Direct band gaps at either the M or S point of the Brillouin zone have been observed for both graphyne and graphyne-3 subjected to different tensile strains. In contrast, graphyne-4 and graphdiyne have been shown to display a direct band gap established at the Γ point, whatever the nature of the implemented strain [122,127]. Subsequent studies by Qui et al. confirmed the effect of biaxial tensile stress in increasing the gap of γ-GDY within the range of 0.47–1.39 eV, while uniaxial tensile strain was found to decrease the band gap to approximately zero at the PW91 level [134].

#### 3.2.2. The Electronic Structure of GDYs

Heteroatom doping is an effective method to alter the band structure of graphyne-like structures. The effect of the functionalization of sp-carbon atoms in γ-GDY on band structure was thoroughly investigated by Koo et al. in 2014 [130]. The hydrogenation of acetylenic linkers increased the band gap from 0.49 to a maximum of 5.11 eV, while fluorination increased the band gap up to 4.5 eV. Figure 14A–C illustrate different configurations for monolayered N-graphdiyne holding different numbers of C and N atoms varying from 24 to 42 entitled C_18_N_6_, C_24_N_4_, and C_36_N_6_, respectively, as reported by Singh in 2019 [144]. A hexagonal lattice was observed for C_18_N_6_ (a = 16.04 Å) and C_36_N_6_ (a = 18.66 Å), while C_24_N_4_ presented a rectangular unit cell (a = 15.97 Å; b = 9.67 Å). All considered crystals displayed semiconducting performance, with band gaps of 2.20 eV (C_18_N_6_), 0.50 eV (C_24_N_4_), and 1.10 eV (C_36_N_6_) [144].

The high-temperature treatment of carbonaceous materials is a conventional method of heteroatom doping. Until now, nitrogen-, sulfur-, and phosphorous-doped GDY derivatives have been prepared by this technique [65,66,67,68,69,145,146]. Chen et al. constructed X-doped graphdiynes by replacing a carbon atom (named C1, C2, and C3) (Figure 14Da) in GDY with heteroatom X, where X = B, N, P, and S. As a consequence, five models for N- (N1, N2, N3, pdN, and NH_2_), three models for each B- (B1, B2, and B3), P- (as P1, P2, and P3), and S- (S1, S2, and S3) doped GDY were constructed. The pdN and NH_2_ models represent the GDY structures doped with pyridinic nitrogen (Figure 14Db) and amino-derived functionalities (Figure 14Dc), respectively. The carbon atoms were divided into three classes that were labeled C1, C2, and C3 according to the arrangement of the structure. Additionally, nine adsorption sites, as indicated by red dots, were investigated in discussed studies.

The doping of pristine GDY with diverse elements (B, N, P, or S) has been found to result in a decrease in the band gap for all analyzed models when using spin-polarized DFT computations and PBE functionals [146]. When the concentration of the dopant was about 1.4 at%, N- and P-doped GDYs were found to be metallic, while doping with boron and sulfur was found to reduce the band gap of starting GDYs from 0.46 to 0.16 and 0.28 eV, respectively. The further decrease in the band gaps was induced by an increase in dopant concentration. B-doped GDYs were found to become metallic after adding 5.6 at% of boron, whereas the band gap of S-doped GDYs decreased to 0.09 eV (Model S3). The planar structure of GDY after doping with B or N atoms was found to be preserved. Due to larger atomic radii of P and S atoms, the out-of-plane distortion of planarity, except for P3 and S3 models, was observed. In terms of the cohesive energies, N, P, and S elements were found to prefer the X2 position, while boron favored the X1 position.

In 2019, Yang et al. theoretically studied the changes in electronic properties caused by the adsorption of H_2_ and O_2_ atoms on GDY [147]. As Figure 14E shows, there were nine possible sites for their adsorption: three top sites (T1, T2, and T3), two hollow sites (H1 and H2), and four bridge sites. It was shown that the most stable adsorption positions are those located at T2 and H2. The adsorption of atoms on the GDY surface is related to the generation of distinguished high adsorption energies (GDY/H = 3.73 eV; GDY/O = 7.53). The latter meant that strong chemical interactions occurred between the adsorbed atoms and the surface. When H atoms were adsorbed, an inadequate 0.1 eV reduction in the band gap was observed. On the contrary, the adsorption of oxygen led to a higher band gap (an increase of 0.2 eV). Furthermore, the introduction of oxygen species to the structure resulted in weak magnetism because of the broken spin degeneration. With the advancement of research with single-layer GYs, researchers have turned their attention to other geometric shapes such as one-dimensional graphyne family members (e.g., nanoribbons and nanotubes) and three-dimensional ones represented by few-layer systems with different stacking arrangements.

**Figure 14 nanomaterials-11-02268-f014:**
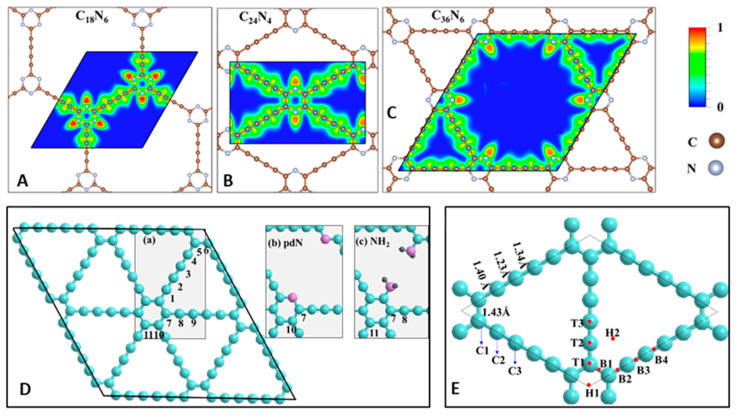
N-graphdiyne nanosheets unit cells of C_18_N_6_ (**A**), C_24_N_4_ (**B**), and C_36_N_6_ (**C**). Reproduced with permission [144]. Copyright 2018, American Chemical Society. (**D**) The available position to be occupied by dopants (Da), pyridinic N-doped GDY (pdN) (Db), and amino-group-doped GDY (NH_2_) (Dc). (**E**) Optimized H_2_ and O_2_ adsorption sites on pristine γ-graphdiyne. Redrawn [146,147].

#### 3.2.3. The Electronic Band Structure of GY Nanoribbons

A nanoribbon (NR), as a 1D derivative of infinite GY sheets, is an example of tuning a band gap by changing the geometry of 2D monolayer GYs. Armchair (AGYNRs) and zig-zag (ZGYNRs) graphyne nanoribbons (differing in widths) were obtained by cutting through a graphyne (or graphdiyne) film along the x and y directions terminated to a benzene ring or acetylene group [140,142]. Gao et al. investigated nanoribbons with benzene rings at their edges. Figure 15 shows AGYNR and ZGYNR arrangements, where n is the number of recurring segments. In contrast to AGYNRs, the n number in ZGYNRs can vary by a half-integer (n + 1/2).

The band gap of the nanoribbon decreased as the width increased because the ribbon tended to revert to a two-dimensional structure; in such a case, the expected band gap of the ribbon was 0.5 eV. The results confirmed that all considered nanoribbons showed semiconductive features, where band gaps were controlled by the widths and the nature of the edge. With the DFT-LDA method, the calculated energy gaps of AGYNR graphyne-based nanoribbons were in the range of 0.59–1.25 eV, while ZGYNRs showed band gaps ranging from 0.75 to 1.32 eV. In the case of graphdiyne nanoribbons (GDYNRs), the energy gaps were in ranges of 0.54–0.97 eV and 0.73–1.65 eV for armchair (AGDYNRs) and zig-zag (ZGYDNRs) configurations, respectively. The effects of the edge arrangement (AGDYNTs and/or ZGYDNRs) of graphdiyne nanotubes (GDYNTs) on band structure were investigated in detail by Shohany et al. [139]. All GDYNTs under investigation exhibited semiconducting behavior, with a fundamental band gap ranging from 0.65 to 0.5 for AGDYNTs and from 0.95 to 0.55 for ZGDYNTs, as summarized in Table 2.

Kang et al. reported that an intersecting electric field could provoke the giant Stark effect in one-dimensional nanostructures, leading to a diminished or even disappeared band gap [148]. In comparison with other band gap modification approaches, the experimental control of an electric field seems to be a much easier way to play with GYs’ band gaps. The effect of an electric field on the band structures of GDYNRs and the band gap decreased as the electric field strength increased due to the strong localization of band-edge states. The band gap decreasing rate was found to be linearly dependent on the ribbon width.

#### 3.2.4. The Electronic Band Structure of Bulky GYs and GDYs

It was shown that 3D graphynes, thanks to the different stack arrangements, could be considered to be both as semiconductors and metals. In 2000, Narita et al. applied the first-principle calculations using a full-potential linear combination of the atomic orbitals method and local-spin-density approximation (LSDA) to optimize the geometry and investigate the electronic properties of one AAA stacking arrangement structure represented as α and three ABA configurations denoted as β1, β2, and β3 of three-dimensional graphyne (Figure 16a) [136].

In Table 3, it was shown that interactions occurring between orbitals 2pπ in α and β3 were greater than those appearing in β1 and β2. As a result, a large split of the π orbitals could be observed, consequently leading to the overlapping of conduction and valence bands in α and β3. The optimized plane lattice constant (a) and bond lengths were nearly the same as in 2D graphyne. The calculated interlayer distance (d) of graphyne (~3.3 Å) was longer than that of graphite (3.17 Å). However, the greater core/core repulsion between neighboring sheets in α and β3 resulted in lower binding energies compared with those found for β1 and β2. Furthermore, the obtained values were approximately 90% of the binding energy of graphite (8.867 eV/atom). This may suggest that graphyne is metastable while it is formulated. In contrast, the more stable nanostructures of β1 and β2 exhibited semiconducting properties. The band gap of β1 was equal to 0.19 eV at the M and L points, whereas the energy gap of β2 was 0.50 eV at the L point. Thus, three-dimensional graphyne was assumed to have layered β1 and β2 organizations and semiconductive features with appropriate energy gaps.

In 2012, Zheng et al. conducted PW91 calculations and showed that the most stable bilayer and trilayer GDYs had their hexagonal rings arranged in the AB (direct Eg = 0.35 eV) and ABA (indirect Eg = 0.33 eV) configurations, respectively [138]. The decline in energy gaps, compared with monolayer species (0.46 eV), was caused by the occurrence of interlayer interactions. The application of an external electric field has been shown to be an effective technique to control the electronic and optical properties of few-layer graphyne-like materials. The two most stable arrangements for bilayer GDY (AB(β1) and AB(β2), owning band gaps of 0.35 and 0.14 eV, respectively) are depicted in Figure 16b. The stable trilayer configurations labeled ABA(γ1), ABC(γ3), and ABC(γ2) were found to present band gaps as high as 0.32, 0.33, and 0.18 eV, respectively (Figure 16c). The less stable AA and AAA configurations of GDY exhibited metallic properties.

In 2013, Nagase et al. employed a vdW-optPBE functional to investigate the relationships between the optical and electronic features of bulky GDY nanomaterials and their configurations. The obtained results were next compared with the results obtained using Heyd-Scuseria-Ernzerhof (HSE06) and LDA functionals [137]. They found that the AA configuration presented the lowest structural stability, accompanied by three AB configurations with energy gaps equal to 0.05, 0.74, and 0.35 eV, respectively. The investigations of Leenaerts et al. on the two-layer α-GY revealed that its band structure was qualitatively different from its single-layer derivative and was affected by the stacking modes of the two layers [149]. The AB staking arrangements exhibited a zero-gap feature similar to the AB configuration of bilayer graphene. It was shown again that electronic properties may be controlled by an applied electric field. The fluorinated GDY exhibited direct semiconductive behavior, with band gaps equal to 2.17 (AB stack-1) and 2.30 eV (AB stack-2), which were more than that of pristine GDY (~0.46 eV) (Figure 16d) [108]. Furthermore, the theoretically estimated band gap of the AB stack-1 configuration was consistent with the experimental value. The experimentally determined band gap of randomly fluorinated triazine-based graphyne by XeF_2_, reported by Szczurek et al., ranged from 3.12 to 3.34 eV and grew with increasing fluorine concentration [150]. An increase in the band gap upon the fluorination of acetylenic bridges is consistent with the decoupling of benzenic chromophores. Multilayer boron-graphdiyne (B-GDY) was comprehensively studied by Li et al. [113]. The calculated band gap energy of the monolayer showed that B-GDY was a direct band gap semiconductor at the Γ point with a value of 1.2 eV, which corresponded well to the experimentally derived band gap (1.1 eV) of the synthetic compound.

### 3.3. Electronic Transport

The carrier mobility of different forms of GYs and GDys was theoretically predicted by Chen et al. In their experiments, the Boltzmann transport equation coupled with the deformation potential theory was applied to a-GY, b-GY, 6,6,12-GY, and GDY, and graphene was used as a reference. The obtained results revealed that almost all GYs and GDYs showed charge mobility values lower by one order of magnitude than graphene. 6,6,12-GY was an exception and presented a higher charge mobility (in the a direction) of around 25% for holes and 37% for electrons than the charge mobility found for graphene. Furthermore 6,6,12-GY presented a tremendous anisotropy of charge mobility along the *a* and *b* axes. The high carrier mobility of 6,6,12-graphyne might be explained by weaker electron–phonon coupling energy and longer relaxation times. Moreover, the rectangular arrangement of the 6,6,12-graphyne framework might cause the anisotropy of charge mobility in the structure [135].

In 2018, Nasri and Fotoohi theoretically investigated the electronic transport characteristics of a device built on N-doped (right electrode) and boron-doped (left electrode) α-armchair graphyne nanoribbons [151]. Four different devices differing in N and B substitution sites (sp or sp^2^) were proposed (Figure 17A). The current–voltage characteristics of the considered systems revealed effective non-linear behavior that led to the generation of a p–n junctions, which, in turn, resulted in rectifying behavior. The rectification properties, however, heavily depended on the deposition site of dopant atoms. The devices with doping atoms substituted on sp^2^ sites (sp^2^–sp and sp^2^–sp^2^) showed a rectification ratio of around ten times lower than those having doping atoms attached to sp sites (sp–sp and sp–sp^2^) measured in the same bias region (Figure 17B). The rectifying behaviors of the described devices may be associated with asymmetric electrode arrangements. The N and B dopants caused crucial variations in electronic structures of α-graphynes, as they generated new sub-bands in valence (VB) and conductive (CB) bands. Those sub-bands seemed to ensure effective conduction and rectification in the described devices. This finding was supported by measurements conducted on pristine α-graphyne nanoribbons. The device built on the latter showed no rectification effect due to the symmetry of both electrodes. The suitability of graphyne and its h-BN (hexagonal boron nitride) derivatives (h-BNynes) to work as a field-effect transistor (FET) was theoretically investigated by Jhon et al. in 2014 [152]. With the aid of non-equilibrium Green’s function combined with the density functional theory (NEGF–DFT) method, they examined the electronic transport of graphene–graphyne–graphene devices by varying graphyne size and carbon chain length.

The concept of such constructed transistors was based on the outstanding electronic mobility of graphene and non-zero band gap graphyne, along with structural/compositional similarities between graphene and graphyne, being robustly connected between graphene electrodes. DFT calculations revealed that both graphyne and h-BNyne-based thin-film transistors (TFTs) showed good on/off ratios on the order of 10^2^–10^3^. Noteworthily, the size of such transistors might be reduced to below 1 nm while maintaining good switching features. Electronic orbital analysis disclosed that, contrary to h-BNynes, electrons in the conduction and valence orbitals were considerably delocalized in graphyne TFTs. The latter finding may suggest that graphyne TFTs could offer more facilitated electron mobilities than h-BNynes.

Single molecule conductance (SMC) was experimentally and theoretically investigated by Li et al. [153]. The measurements were performed using a scanning tunneling microscope, where the α-graphyne unit was chemically attached to the Au electrode and the STM tip. Amine (NH_2_) functionalities were used as an anchoring agent due to their ability to create molecular joints with negligible conductance aberrations (Figure 17C). The measurements conducted on α-graphyne and hexabenzocoronene representing the graphene molecular unit revealed that the α-graphyne units showed much higher conductance (106 nS, 1.9 nm) than the shorter units (1.4 nm), thus potentially being more conductive than hexabenzocoronene (14 nS). This unusual behavior originated from the electronic structure of both compounds. It turned out that the α-graphyne units had smaller HOMO-LUMO gaps than the graphene molecular units, so the transmission through the α-graphyne core was higher. The high molecular conductance of α-graphyne molecules also came from their rigid and planar structure. The measurements revealed that carbo-butadiene wires showed a single molecular conductance that was 40 times smaller than that found for α-graphyne. The reason for this feature lies in the flexibility of the n-p-conjugated DBA framework, capable of adopting diverse geometries. The transport properties of different conformations of DBA molecules carried out with the NEGF-DFT model were strongly influenced by the measure of the rotation angle around the –C–C=C–C– sites. The increase in twist angle resulted in a higher HOMO-LUMO gap and thus a lower transmission between them. Finally, it was shown that the α-graphyne-based device showed excellent gating properties, understood as an increment of the SMC with increasing negativity of the gating potential (Figure 17D). Furthermore, the on/off ratio found for α-graphyne had an order of magnitude of ~15.

### 3.4. Optoelectronic Properties

The optical features of N- and B-doped graphynes were theoretically studied by Bhattacharya and Sarkar [154]. The investigated structures presented, similar to pristine graphyne, optical anisotropy independent to the direction of applied electric field (Figure 17A, first column). The authors found that below an energy of 0.4 eV, the optical response was governed by the intra-band shift coming from free charges (Figure 18A, second column). The analysis of the static dielectric tensors revealed that doped graphynes showed a better electric conductivity and higher mobility of charges than pristine GY, creating the opportunities for their application in optoelectronics. The spin-polarized optoelectronic properties of α-graphyne were theoretically investigated by Yang et al. using NEGF–DFT [155]. They showed that photocurrents were generated by irradiating investigated devices with different light wavelengths (from UV to IR), and the polarization of the formed photocurrents strictly depended on the type of contact applied. Both M1 and M2 devices (Figure 18B) produced spin-down photocurrents, whereas the M2 device could also generate spin-up ones (Figure 18C). It was revealed that photocurrents might also be guided by reversing the electrodes’ magnetization. Generally speaking, two spatially separated spin photocurrents appeared on the antiparallel polarized electrodes. Functionalization is another approach to manage the optoelectronic characteristics of GY and GDY nanostructures. Theoptoelectronic application of Gdy:ZnO nanocomposites was experimentally investigated by Jin et al. [156] Their experiment involved measuring I–V characteristics in the dark and under the influence of UV radiation. When the samples were illuminated, an increase in current due to light absorption was observed. Chronoamperometry was employed to record the rise/decay time of the current measured without and with light irradiation. The obtained results revealed that the current changes strongly depended on the photochemical character of investigated devices. The junction created between the GDY and ZnO nanoparticles strongly enhanced the charge transfer between these components and, consequently, the photoresponse. The GDY:ZnO/ZnO system manifested an excellent optical response of 1260 A W^−1^, with a rise/decay time as short as 6.1/2.1 s. The device built only on zinc oxide nanoparticles held a responsivity of 174 A W^−1^ and a much more extended rise/decay time (32.1/28.7 s). Li et al. proposed deep UV photodetectors (Figure 18D) based on TiO2:GDY nanocomposites [157]. The published results proved that the considered composites were suitable as UV photodetectors, disclosing an outstanding optical response of 76 mAW^−1^ and a rise/decay time of 3.5/2.7 s. The cited examples show the promising and efficient properties of graphyne derivatives. Their high charge mobility, on/off ratio, and photoresponse makes them prospective materials for constructing nanoscale electronics and optoelectronics.

It is noteworthy that metal oxide dopants in GDY nanocomposites, the optoelectronic amplifiers, do not seem to be mandatory additives. Zhang et al. constructed metal-free, flexible photodetectors built on GDY:PET composites. Those built devices were characterized by excellent mechanical, electronic, and optoelectronic properties. The experimentally investigated responsivity and photocurrent reached outstanding values of 1086.96 μA W^−1^ and 5.98 μA cm^−2^, respectively. Moreover, the considered composites showed good photoresponses, even after undergoing 1000 bending and twisting cycles. The recorded loss in photocurrent was 25.6% (bending force) and 35% (twisting force) [158].

## 4. Magnetism of Pure and Doped Graphyne-Like Materials

### 4.1. Theoretically Investigated Magnetic Properties

Local defects and substituents in carbonaceous materials might provoke superconducting or ferromagnetic features that can even appear at ambient temperature. Until now, magnetism in carbon has been experimentally revealed for (i) interacting radicals, (ii) carbons with a mixed hybridization (sp^2^ + sp^3^), (iii) amorphous carbonaceous materials doped with trivalent atoms (P, N, or B), (iv) diverse nanostructures (graphite, diamond, and foams), and (v) fullerenes [159,160,161,162]. Apart from that, vicinity to metallic ferromagnets or the contamination of transition metals (Fe, Cr, or V) has been shown to generate spin polarization in carbon structures [159,160,162]. Localized magnetism and the zig-zag rib of graphene nanoribbons or neighboring vacancies and dangling bonds in pure carbon structure are representative of this concept. The theoretical considerations and fruitful synthesis of monolayer graphyne have led to considerable interest in research on the magnetic properties of pure or/and doped graphyne structures.

#### 4.1.1. Metal-Doped GYs

The magnetic features found in materials with electrons occupying only *s* and *p* orbitals, rather than the traditional *d* or *f* ones, might be remarkably appealing to spintronic applications. However, the source of magnetism in pure carbonaceous materials is not fully understood. In this section, we report works addressing the magnetic properties of GYs. The effect of dopant distribution and functional groups on adjusting the electronic or magnetic properties of GYs might lead to new promising electronic, optoelectronic, and spintronic devices. The calculations revealed that unmodified γ-graphyne is a nonmagnetic semiconductor [128], while the adsorption of transition metal (TM) atoms might drastically change the electronic structure of and add ferromagnetic features to GY nanostructures [163,164]. In 2012, He et al. theoretically investigated (DFT + U) the electronic structure and magnetism of GDY and GY doped with single 3*d* transition metals (V, Cr, Mn, Fe, Co, and Ni) [164] The adsorption of metal atoms on the GDY and GY surfaces generated charge transfer between metallic adatoms and polymeric sheets. Moreover, the adsorption might generate electron redistribution in the *s*, *p*, and *d* orbitals of transition metal atoms. Except for vanadium, the mentioned factors caused decreases in the magnetic moments of adsorbed metals (Cr, Mn, Fe, and Co). Additionally, they were ranked as follows: Cr > Mn > V > Fe > Co for TM-GDY and Cr > V > Mn > Fe > Co for TM-GY; see Figure 19A. The energy of spin polarization (ΔE_spin_), taken as the difference between the nonmagnetic and magnetic states, was higher than 1.1 eV, suggesting the substantial stability of the spin-polarized states of transition for metal-doped GDY and GY nanostructures.

In 2017, Lee et al. applied DFT calculations to investigate the doping efficiency of the transition metals of 3*d*, 4*d*, and 5*d* groups deposited on a γ-GY surface [165]. For this purpose, different high-symmetry adsorption sites such as top (T), bridge (B), and hollow (H) were chosen, as depicted in Figure 19B. It was shown that adatoms typically occupy the H1 sites over ethynyl rings of γ-graphyne (taking Fe (μB: ~2.08) as an example). The lower atomic radii of Co (μB: ~1), Ni (μB: ~0), and Cu (μB: ~0), as well as Re (μB: ~1), caused those dopant atoms to be preferably placed at the H3 sites of the acetylenic rings. Magnetic moments of metals of the 4*f* group were found to be higher in comparison with 3*d* transition metals, and lanthanides with sufficiently large atomic radii are good candidates to be introduced in GY rings. Ren et al. used comprehensive first-principle calculations to study the magnetic properties of *β*-GY doped with different rare-earth (RE) atoms (La, Ce, Pr, Nd, Pm, Sm, and Eu) [166]. The β-GY was found to undergo a transition from semiconductor to metal. The introduction of external atoms such as neodymium, promethium, samarium, and europium (local magnetic moments in the range of 4.1–7.3) were able to translate into higher values of magnetic moments for metal-graphyne complexes (>4.1 μB). As expected, the carbon atoms neighboring dopant atoms were found to have a modest contribution to generated magnetic moments. In 2014, Alaei et al. studied two zig-zag graphyne nanotubes (ZGYNTs) and two armchair graphyne nanotubes (AGYNTs) doped with iron, cobalt, and nickel [167]. It was shown that a 12-membered ring (12-C), a hollow site surrounded by acetylenic linkers in GY, was the most preferable (the most stable) site for the deposition of those metals. The adatoms nested in the plane of the ethynyl rings and formed bonds with adjacent carbons. Complexes of Fe (μB: ~2.06) and Co (μB: ~1) with different GYNTs were magnetic and showed many features typical of metals, semimetals, half semimetals, and half-semiconductors [168]. Ni complexes (μB: ~0.01) were found to be nonmagnetic semiconductors exhibiting energy gaps narrower than those found for starting nanotubes (Figure 19A).

#### 4.1.2. Non-Metal Doped GYs and GDYs

The electronic properties and magnetism of GYs can be also driven by applying diverse non-metal doping agents. In 2014, Drogar et al. stated that widening the band gap (~2 eV) and provoking a magnetic moment (~1 μB) of α-GY could be realized via the simple hydrogenation reaction of the latter due to the cleavage of π-bonds and the creation of unpaired electrons [169]. Subsequently, in 2018, Wang et al. studied the tuning of mono- and bilayer GY features after the hydrogenation of different carbon atoms in GY [170]. Unlike the distribution of μB in TM-GY, in which the magnetism derives from the d- or f-electrons, the unpaired 2p electrons essentially provide the magnetic moments at the non-hydrogenated carbons of GY. As indicated in Figure 19C, there are two sites (C1 (T) and C2 (B)) on which hydrogenation can occur. Twelve hydrogen atoms are located at Ti sites in the aromatic ring and the B1 sites in acetylenic linkers of GY nanosheets. It was observed that the magnetic moment of the monolayer reached a maximal value (1.59 μB) for three hydrogen atoms, whereas the magnetic moments progressively declined to 0 μB when the number of hydrogen atoms varied from 4 to 6. These results suggest that the hydrogenation of half C atoms (sp- or/and sp^2^-hybridized) may result in a maximal magnetic moment, similarly to what was found for graphene [171]. Theoretical calculations of the total magnetic moments of bilayers with different stacking arrangements yielded values larger than 1.52 μB. DFT is the most frequently utilized computation method to investigate the electronic properties and magnetism of single-layer graphdiyne (GDY) doped with non-metallic elements, such as boron, nitrogen, oxygen, phosphorous, or sulfur [172]. The position of dopant atoms has a profound impact on the relevant characteristics of GDYs. Three possible places for doping are aromatic carbons (X-b) and two carbons in ethynyl linkers (X-1 and X-2), depicted in Figure 19D. Considering the cohesion energies found for the studied structures, one can conclude that boron and sulfur prefer to exchange the sp^2^ aromatic carbons. In turn, the N, O, and P atoms favor substituting the carbons in the ethynyl bridges. It was shown that nitrogen atoms deposited on both GDYs and GY monolayers did not modify their magnetic properties, and all of them remained nonmagnetic [154]. In contrast, GDY structures doped with B, O, P, or S (structures X-1 and X-2) deposited at acetylenic chains offered spin polarization and were magnetic (μB varying from 0.31 to 1.26 depending on the type of dopant and applied functionals). The deposition of these elements at the aromatic site did not alter magnetic features, and doped GDYs were nonmagnetic.

**Figure 19 nanomaterials-11-02268-f019:**
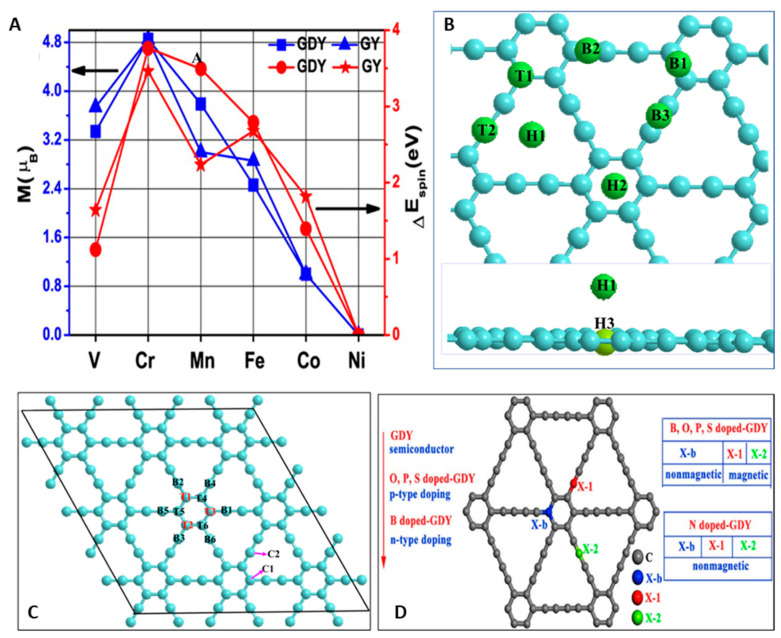
The magnetic moment (μB) and ΔE_spin_ (**A**). Reproduced with permission [164] Copyright 2012, American Chemical Society. The adsorption sites (green spheres) of transition metal atoms on a γ-graphyne film seen from the top (upper image); the in-plane case of H3 and its relation to the H1 site seen from a side (bottom image), blue points represent carbons (**B**). Redrawn [165]. Possible sites on GY for hydrogenation (**C**). Redrawn based on [171]. Possible site for doping by X = B, N, O, P, and S atoms (**D**). Reproduced with permission [172]. Copyright 2020, Elsevier.

### 4.2. Experimentally Investigated Magnetic Properties

Following comprehensive theoretical studies on the magnetic properties of modified GYs, efforts have been directed to experimentally explore and measure magnetism in GYs [173]. In 2017, Huang et al. investigated the impact of N doping on the paramagnetic properties of GDYs and demonstrated the crucial role of nitrogen atoms deposited on the benzene ring in building the local magnetic moment [174]. Based on the M-H curves (Figure 20A,B) measured for GDY and N-GDY at 2 K, the authors showed that N-GDY and GDY presented low magnetization values of 0.96 and 0.51 emu/g, respectively. The magnetization obtained for N-GDY containing 5.29% of N atoms was on the same level as the value found for fluorinated RGO [175]. These results contradicted recent theoretical findings, which claimed that N in a chain or ring is non-paramagnetic. As such, more research is required to find the origination of paramagnetism [154,172].

The spin-polarized calculations proved that a distinct local magnetic moment (μ_B_ = 0.98) originates from nanostructures with deposited asymmetric pyridinic nitrogen (Py-1N), seen in Figure 20C,D. The structures bearing symmetric pyridinic nitrogen substitution (Py-2N) or N atoms attached to ethynyl linkers appeared to be nonmagnetic (Figure 20E,F). Finally, the investigated systems did not show any ordered ferromagnetic or ferrimagnetic properties.

Further research aiming to discover the relationship between doped heteroatoms and magnetization in GY structures continued. In 2017, Zheng et al. [176] reported that, contrary to pyridinic-N, the vacancy, carbonyl, and hydroxyl functionalities of GDY contribute to the magnetic properties of thermally treated GDYs. Furthermore, DFT calculations indicated that the OH groups at the chain of the GDY layer are a considerable source of unpaired electrons and may favor antiferromagnetism in annealed GDY. Moreover, the annealing of GDY at 600 °C (GDY-600) may lead to complex magnetic properties, depending on the applied measurement temperatures. Paramagnetic characteristics could be noticed below 50 K, and a hump seen in the range of 50–200 K demonstrated that both paramagnetic and antiferromagnetic phases coexist in the considered materials.

In 2019, Huang et al. investigated the effect of sulfurization on the induction of ferromagnetic characteristics into GDYs. They found that S-doped GDY presented strong residual magnetization (>0.047 emu/g) at ambient temperatures. The investigated systems were characterized by the transition temperature being close to 460 K [177]. The local magnetic moment and electron interactions occurring between C and S atoms were found to be responsible for the appearance of the ordered internal ferromagnetism in the investigated materials. The ferromagnetic behavior was also confirmed by magnetic hysteresis (M-H) loop measurements with different temperatures. In order to investigate the influence of sulfur doping on magnetic properties of GDY, temperature-dependent magnetic susceptibility χ-T curves (the applied magnetic field H = 500 Oe) and magnetization M-H curves for GDY350 and S-GDY by VSM were measured (Figure 21a–d) [177]. These findings are very promising in terms of S-doped GDY suitability in magnetic storage devices.

## 5. Mechanical Properties of Graphynes

Graphene with an intrinsic tensile strength of 130 GPa, and Young’s modulus equal to 1TPa is recognized as the strongest material ever tested [178]. Zhang et al. theoretically investigated mechanical properties of different forms of GYs (α-, β-, γ-, and 6,6,12-GYs), as well as graphene used as a reference [179]. The tensile stress (~125 GPa) and Young’s modulus (0.99 TPa) obtained by molecular dynamic calculations confirmed the extraordinary mechanical resistance of graphene. In contrast, all considered graphynes presented 50–70% lower mechanical strength than graphene, whereas the decline of tensile stress and Young’s modulus of graphynes was strictly related to an increasing amount of acetylenic linkage. A similar dependence of elastic properties on the proportion of acetylenic bridges in α-, β-, γ-graphynes was also observed by Hou et al. in 2014 [123]. Noteworthy, a deviation from the hexagonal structure of graphene and graphynes, as it happens for 6,6,12-graphyne, resulted in the appearance of anisotropy of mechanical properties measured along with x (zig-zag) and y (armchair) directions [179,180]. The anisotropy in mechanical and electronic properties makes 6,6,12-graphyne useful in diverse potential applications.5.

## 6. Conclusions and Outlook

Synthetic methods for the preparation of graphyne-like structures doped with heteroatoms and controllable size and dimensions were collected and summarized. Doping (i.e., the replacement of carbon atoms or covalent bond formation with foreign atoms) methods such as heating and annealing techniques have crucial drawbacks in the preparation of carbon materials; uncertainty in the values and position of heteroatoms, along with the destruction of the intrinsic properties of pristine materials, are inevitable disadvantages. To avoid the abovementioned flaws, engineering the surface of GYs to include desired heteroatom is necessary. Different methodologies have been successfully applied with high reproducibility to prepare a wide range of GYs with ordered structures in which heteroatom occupy predictable positions on the surface. The adjustment of morphology (such as 1D, 2D, and 3D) and composition (doping with N, P, F, S, Cl, H, O, B, etc.) are influential approaches to modulate the band structures and (subsequently) electric, optical, and magnetic properties of graphyne-like structures. The size- and composition-dependent band structure, electronic transport, spin-polarized optoelectronic properties, or photosensitivity, and appearing magnetic moment of these allotropes of carbon material have been investigated. There is an understanding that research into the magnetic properties of carbon materials is still in its infancy stage, and there is a substantial gap between calculations and experimental findings. Though this topic needs many more firm, particularly experimental discoveries, it is already clear that this type of material could be exploited in spintronics technology. Benefiting from the high specific capacity, remarkable cycle performance, and inflated adsorption capacity of ions and gases, GYs could also have a promising future for applications in electrochemical energy storage such as batteries, capacitors, and hydrogen fuel cells. Theoretical research has proved GY’s usefulness for not only understanding but also predicting the structures and properties of new modified systems as well as for preselecting those that most merit experimental study.

## Figures and Tables

**Figure 1 nanomaterials-11-02268-f001:**
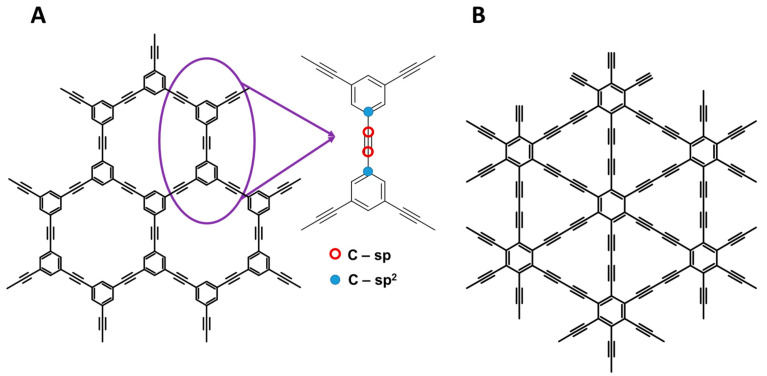
The structure of graphyne with indicated distribution of sp (red rings) and sp^2^ (blue circles) hybridized carbons (**A**); The representation of graphdiyne structure (**B**).

**Figure 2 nanomaterials-11-02268-f002:**
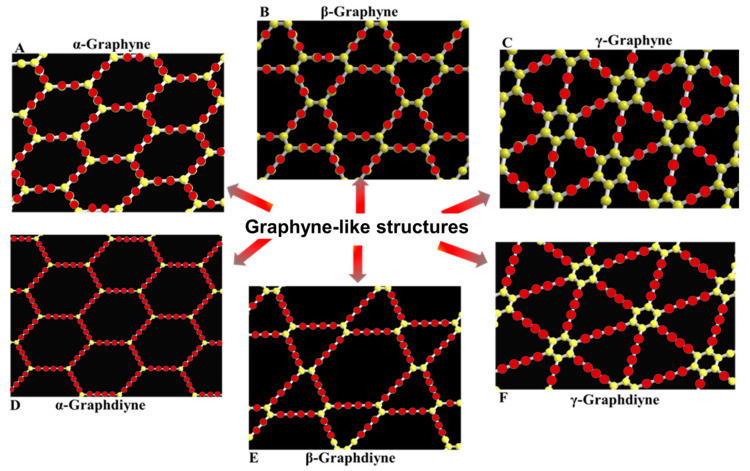
Representation of different graphyne and graphdiyne–like structures. α-graphyne (**A**); β-graphyne (**B**); γ-grapyne (**C**); α-graphdiyne (**D**); β-graphdiyne (**E**); γ-graphdiyne (**F**). The red dots represent acetylenic (graphynes) and diacetylenic (graphdiynes) bridges (Csp) and yellow ones—sp^2^ hybridized carbons.

**Figure 3 nanomaterials-11-02268-f003:**
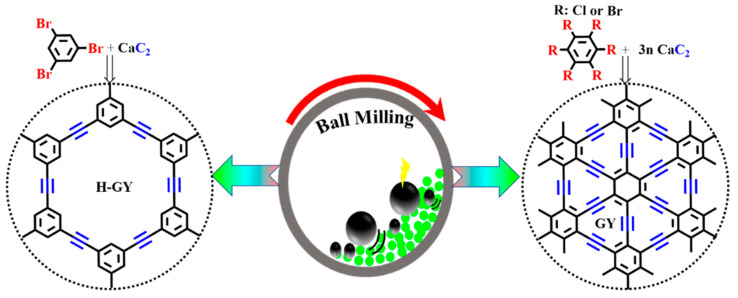
Schematic representation of mechanochemical synthesis for preparation of GY and its derivatives. Redrawn [60,61].

**Figure 4 nanomaterials-11-02268-f004:**
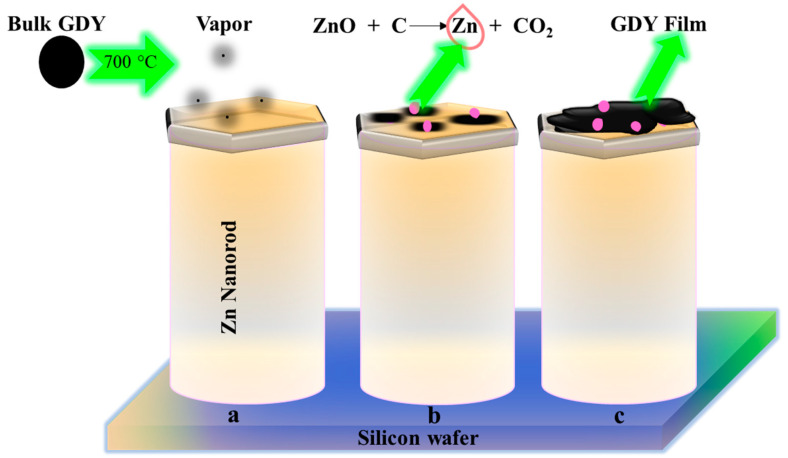
The illustration of the increase in a GDY layer in the VLS process on the head of a single ZnO NR (**a**) ZnO NRs in the presence of GDY vapors, (**b**) Zn droplet on the head of the ZnO NRs, and (**c**) Zn droplets scattered in GDY thin film, produced by connected several neighboring flakes of GDY. Redrawn [62].

**Figure 5 nanomaterials-11-02268-f005:**
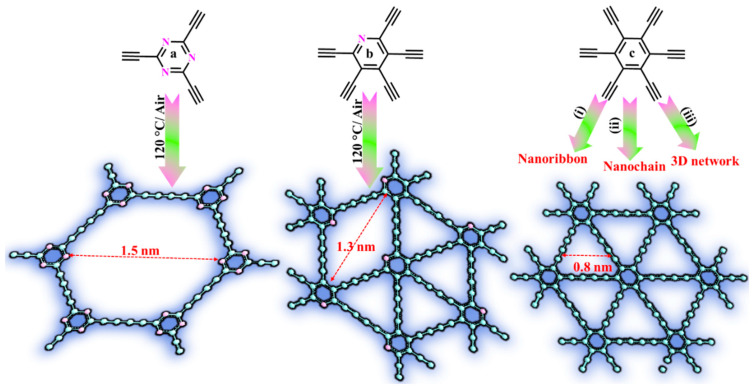
Thermal treatments of (**a**) TET, (**b**) PEP, and (**c**) HEB: (i) gradually heated to 120 °C/N_2_; (ii) rapidly heated to 120 °C/Air; and (iii) gradually heated to 120 °C/Air. Redrawn [63,64].

**Figure 6 nanomaterials-11-02268-f006:**
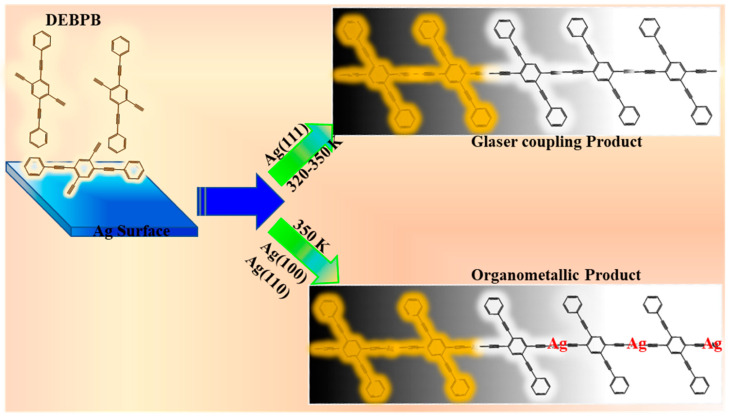
Schematic representation of the DEBPB coupling (Glaser reaction) carried out on silver surfaces. Redrawn [75].

**Figure 7 nanomaterials-11-02268-f007:**
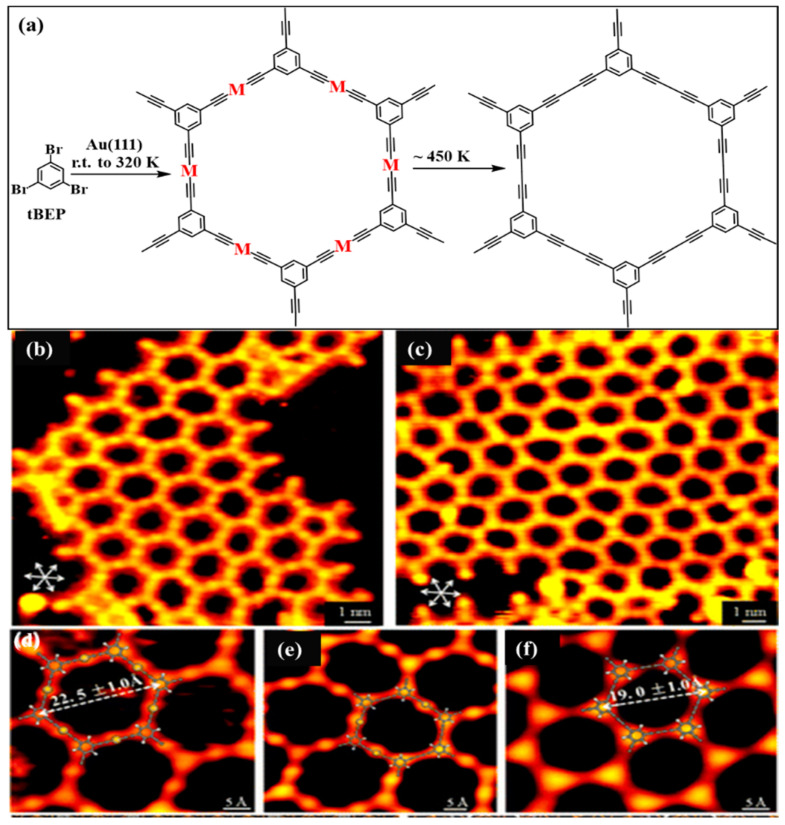
Schematic representation of the growth of the organometallic framework after the deposition of tribromophenyl on Au(111) surface (from RT to 320 K) and the formation of GDY after heat treatment at 450 K (**a**). Scanning tunneling microscopy micrograph revealing the creation of GDY before and after heat treatment at 450 K (**b**,**c**). Detailed STM pictures of the C–Au–C framework (**d**), the mixture of C–Au–C networks and GDY fragments (**e**), and the GDY layer (**f**). The modeled structures of all considered networks are superimposed on the STM micrographs. Reproduced with permission [80]. Copyright 2016, American Chemical Society.

**Figure 8 nanomaterials-11-02268-f008:**
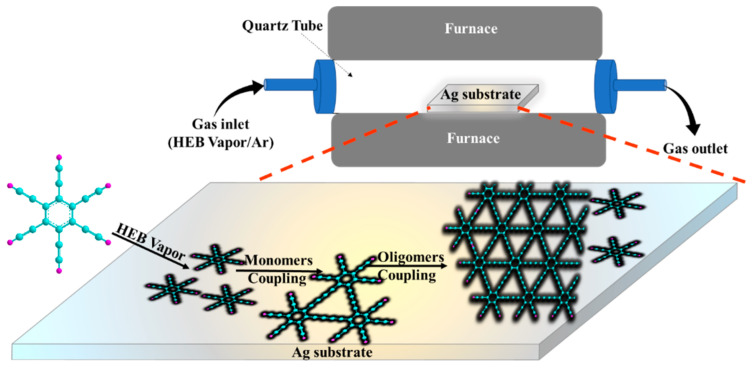
The formation of the GDY sheets on the silver surface with the aid of the CVD technique. Redrawn [86].

**Figure 11 nanomaterials-11-02268-f011:**
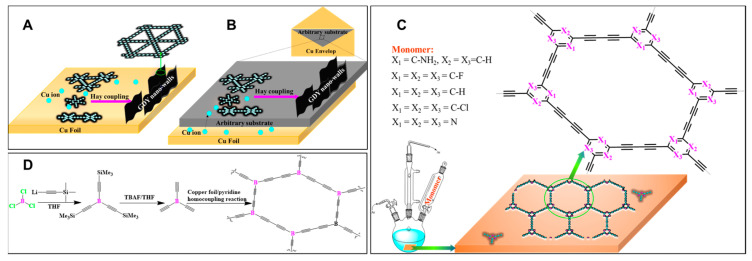
Proposed reaction process of GDY nanowall formation on Cu foil (**A**); envelope strategy for preparation GDY nanowalls on arbitrary substrates (**B**). Redrawn [35,106]; Glaser homocoupling reaction on Cu foil (liquid/solid method) (**C**). Redrawn [107,108,109,110,111,112]; boron-graphdiyne (B-GDY) preparation through homocoupling reaction on Cu foil (solid/liquid method) (**D**). Redrawn [113].

**Figure 13 nanomaterials-11-02268-f013:**
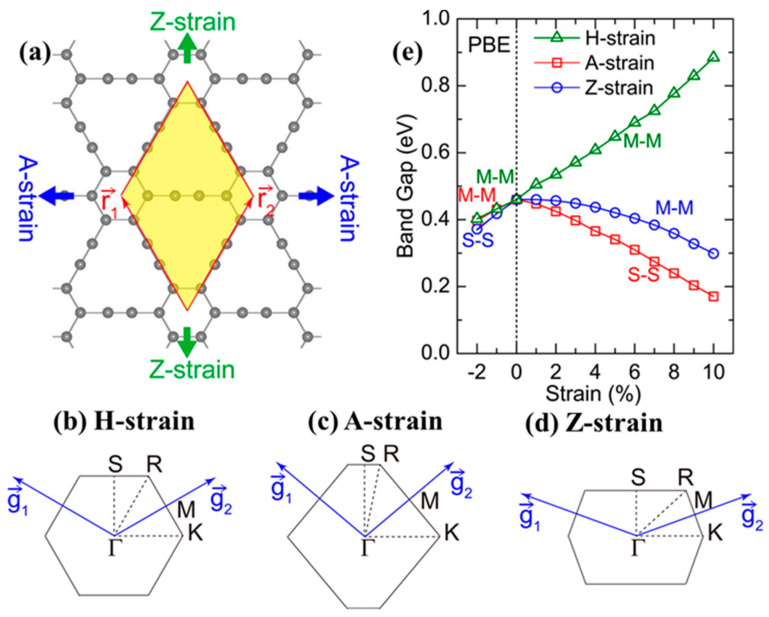
The structure of GY film. The yellow parallelogram indicates the unit cell (**a**). Brillouin zone with high-symmetry points marked beneath H-strain (**b**), A-strain (**c**), and Z-strain (**d**). The band gap shift under different applied strains determined from GGA-PBE (**e**). Reproduced with permission [127]. Copyright 2013, American Chemical Society.

**Figure 15 nanomaterials-11-02268-f015:**
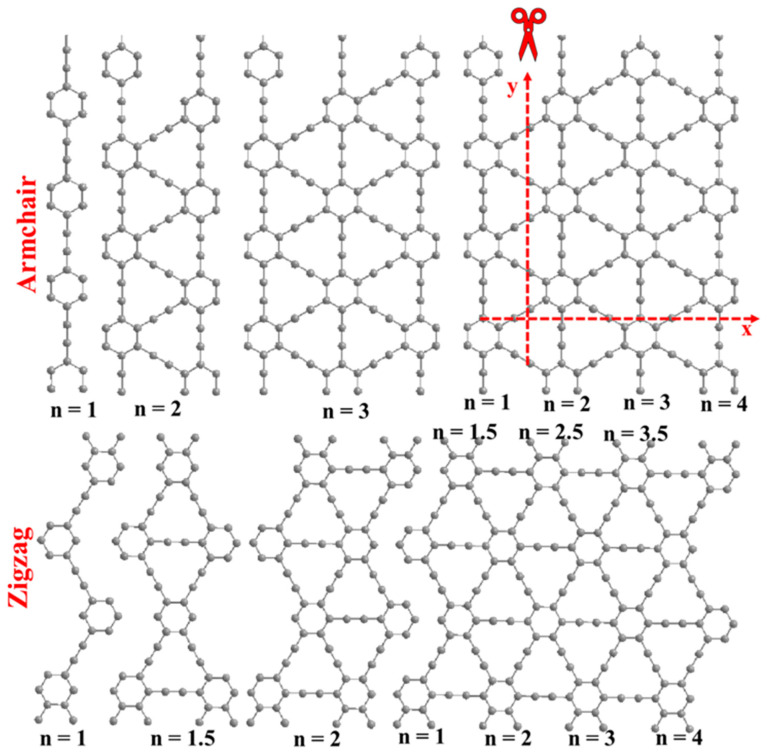
Armchair and zig-zag GY nanoribbons were achieved through the scissoring of the layer along the x and y directions. Redrawn [140].

**Figure 16 nanomaterials-11-02268-f016:**
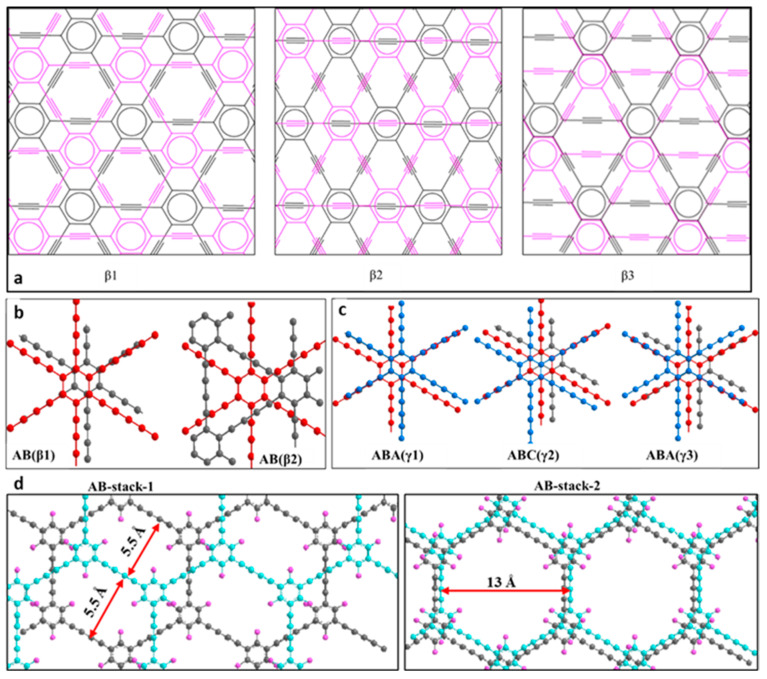
ABA stacking arrangements of 3D graphyne. The A and B sheets are represented by grey and pink structures (**a**). Redrawn [136]. (**b**) Optimized arrangements of double-layer GDY labeled AB(β1) and AB(β2) (red top layer and grey bottom layer, respectively); (**c**) three potential forms of the trilayer GDYs: ABA(γ1), ABC(γ2), and ABC(γ3) arrangements (blue top layer, red middle layer, and grey bottom layer). Redrawn [138]. (**d**) AB-1, AB-2, and AB-3 represent structures of bulk GDY. Red arrows show the directions of the in-plane shift of two sheets in the cell. Redrawn [108].

**Figure 17 nanomaterials-11-02268-f017:**
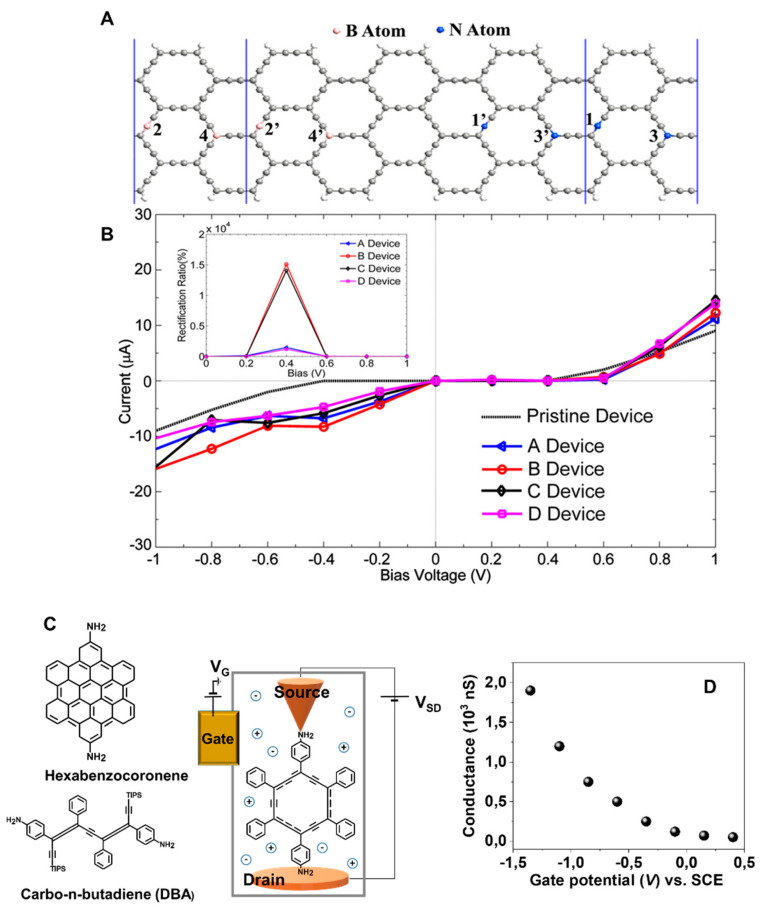
The scheme of devices built on an armchair graphyne nanoribbon (AGyNR) doped with boron and nitrogen atoms (**A**). The I-V curves of doped devices confronted with a pristine system; the inset figure shows the rectification ratio, RR (V), of all doped systems (**B**). Reproduced with permission [151]. Copyright 2018, Elsevier. The schemes of STM devices designed for the investigation SMC of carbobenzene, hexabenzocoronene, and carbo-n-butadiene (DBA). The molecules were attached via NH_2_ linkers to gold STM electrodes; TIPS: triisopropylsilyl (**C**). The single molecule conductance (SMC) and gate potential (VG) relationship found for the carbo-benzene sample. In this experiment trihexyl tetradecyl-phosphonium-bis(2,4,4-trimethylphenyl)phosphinate) was applied as the electrochemical gating electrolyte and the bias voltage was constant (0.1 V); SCE: saturated calomel electrode (**D**). Redrawn [153].

**Figure 18 nanomaterials-11-02268-f018:**
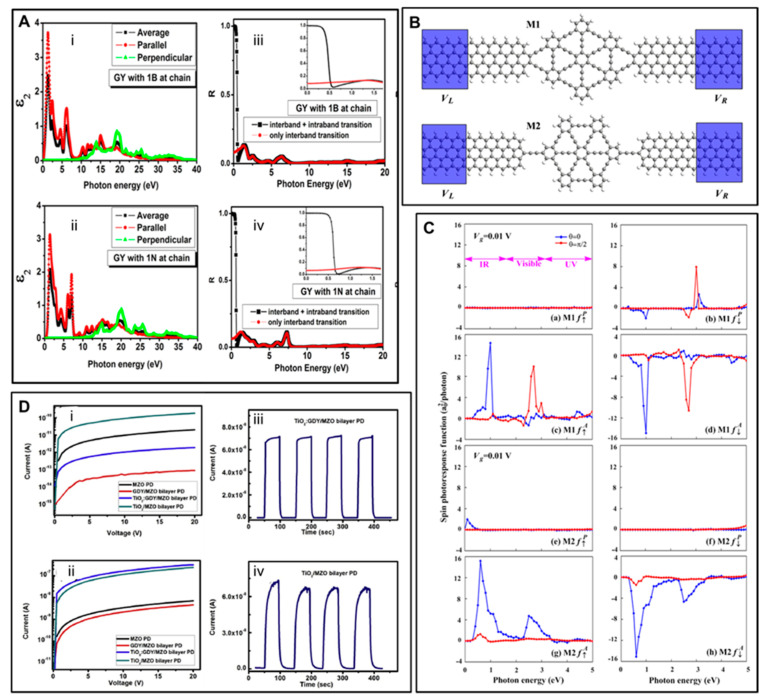
The 1B (**Ai**) and 1N (**Aii**) doped GYs’ relationships between the imaginary part of the dielectric function and photon energies taken for different directions of electric fields. The reflectivities of inter- and intraband transitions found for 1B (**Aiii**) and 1N (**Aiv**) doped GYs. Reproduced with permission [154]. Copyright 2016, American Chemical Society. Scheme of the designed devices (**B**). The parallel and antiparallel spin-polarized photoresponse of M1 (**Ca–d**) and M2 (**Ce-h**) devices. Reproduced with permission [155]. Copyright 2017, IOP Publishing Ltd. I–V sweeps of the considered photodetectors working without (**Di**) and (**Dii**) with UV light (254 nm) irradiation. The on/off features of the TiO2:GDY/MZO bilayer PD (**Diii**) and the TiO2/MZO bilayer PD (**Div**). Reproduced with permission [157]. Copyright 2020, Elsevier.

**Figure 20 nanomaterials-11-02268-f020:**
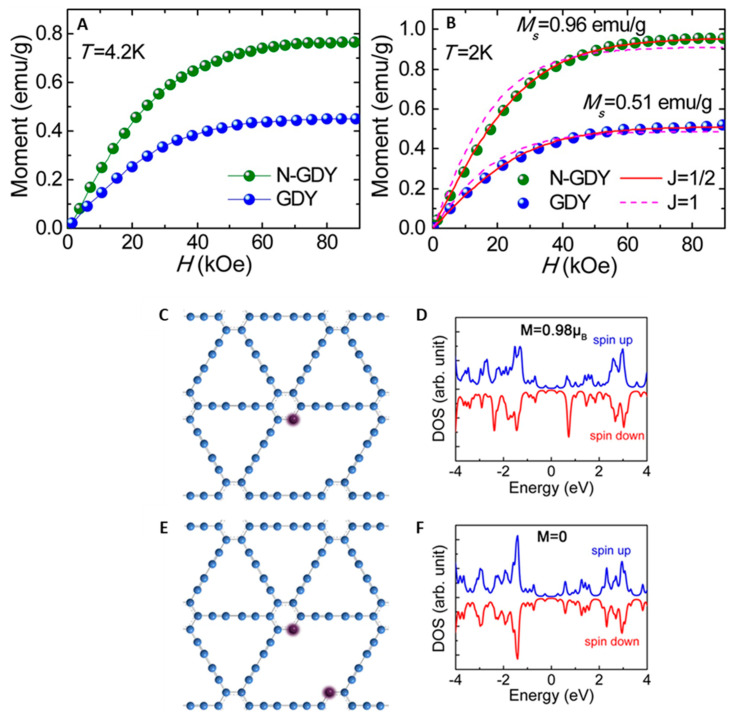
M-H curves obtained at 4.2 K (**A**) and 2 K (**B**) for GDY and N-GDY. Obtained results were fitted to the Brillouin function with J = 1/2 (solid line) and J = 1 (dashed line) (**B**). Spin-resolved DOS of N-GDY films with nitrogen atoms deposited on the benzene ring in GDY. The upper panel (**C**,**D**) shows the paramagnetic S = ½ system, while the bottom one the antiferromagnetic one (**E**,**F**). Reproduced with permission [174]. Copyright 2017, Springer Nature.

**Figure 21 nanomaterials-11-02268-f021:**
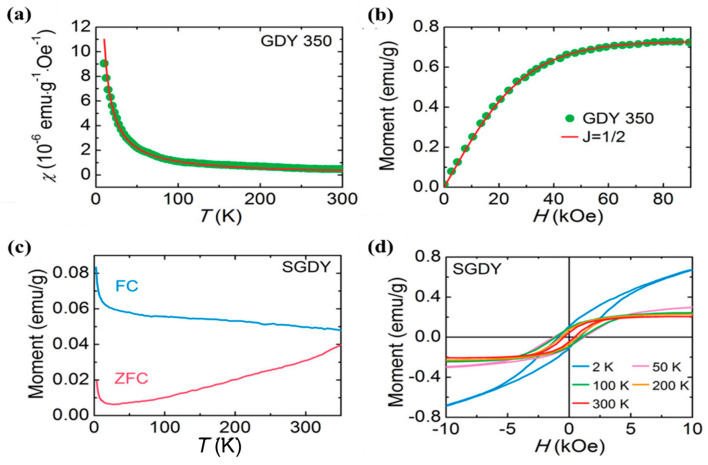
The χ-T relationship examined in a temperature range of 2–300 K (**a**) and an M-H curve taken at 2 K for GDY 350 (**b**). Temperature–magnetization relationship, where FC is field cooled mode and ZFC–zero field cooled mode (**c**) and hysteresis loops recorded for S-doped GDY (**d**). Reproduced with permission [177]. Copyright 2019, American Chemical Society.

**Table 1 nanomaterials-11-02268-t001:** Synthetic methods employed in cross-coupling or homocoupling reactions for the preparation of graphyne derivatives; year when a given method was invented is indicated.

Synthetic Method	Year	Condition	Reaction	[Ref.]
Glaser coupling	1869	− Catalyst: Cu(I) − Oxidant: O_2_ − Base: ammonia − Terminal alkyne	2 R–C≡C–H → R–C≡C–C≡C–R	[43]
Eglinton coupling	1959	− Catalyst: Cu(II) − Base: pyridine − Terminal alkyne	2 R–C≡C–H → R–C≡C–C≡C–R	[44]
Hay coupling	1962	− Catalyst: Cu(I) − Base: tetramethylethylenediamine (TMEDA) − Terminal alkyne	2 R–C≡C–H → R–C≡C–C≡C–R	[45]
Negishi cross-coupling reaction	1977	− Catalyst: palladium (0) − Nickel: co-catalyst- − Organic halides or triflates	RX + R′-ZnX′ + ML_n_ → R-R′ X = Cl, Br, I, triflate and acetyloxy X′ = Cl, Br, I R = alkenyl, aryl, allyl, alkynyl or propargyl R′ = alkenyl, aryl, allyl, alkyl, benzyl, homoallyl, and homopropargyl. L = triphenylphosphine, DPPE, BINAP or chiraphos M = Ni, Pd	[46]
Hiyama coupling	1988	− Palladium (0)	R-SiR′′_3_ + R′-X → R-R′ R: aryl, alkenyl or alkynyl R′: aryl, alkenyl, alkynyl or alkyl R′′: Cl, F or alkyl X: Cl, Br, I or OTf	[47]
Sonogashira cross-coupling reaction	2002	− Aqueous media − Mild base − Palladium (0) − Copper as co-catalyst	R_1_-X + H–C≡C–R_2_ → R_1_–C≡C–R_2_ R1: aryl R2: aryl or vinyl X: I, Br, Cl or OTf	[48]

**Table 2 nanomaterials-11-02268-t002:** Calculated band gaps of GYs based on different summarized methods.

Name/Percentage of Acetylenic Linkages %	Band Gap/Method	[Ref.]	Name	Band Gap/Method	[Ref.]
*γ*-GY	0.447/PBE 0.448/PBE-D 0.45/PBE 1.2/MNDO 0.52/FP-LCAO 0.47/PBE 2.23/B3LYP 0.46/PBE 0.46/PBE 0.96/HSE06 0.94 HSE06 0.474 0.454/PBE	[122] [122] [124] [33] [121] [125] [125] [126] [127] [128] [127] [128] [129]	*γ*-GDY	0.5/PBE 0.44/LDA 1.10/GW 0.53/FP-LCAO 1.22/HSE06 0.46/PBE 0.52/PBE 1.18/B3LYP 0.89/HSE06 0.47/PW91 0.9/HSE06 1.21/HSE06 0.485/PBE	[130] [131] [131] [121] [132] [133] [125] [125] [127] [134] [134] [106] [129]
*β*-GY	0.028/PBE 0.04/PBE-D	[122] [122]	*α*-GY	0/PBE 0.005/PBE-D	[122] [122]
6,6,12-GY	0/PBE	[135] ^a^	Graphyne-3	0.6 at M/FP-LCAO 0.56/PBE 0.566/PBE	[121] ^b^ [127] [129]
Graphyne-4	0.59/FP-LCAO 0.54/PBE 0.542/PBE	[121] ^c^ [127] [129]	Bulk-GY	0–0.5/FP-LSDA	[136] ^d^
Bulk-GDY	0.05–0.74/HSE06	[137] ^d^	Trilayer GDYs	0.18–0.33/PW91 0.9/HSE06	[138] ^d^ [106]
Bilayer GDYs	0.14–0.35/PW91 0.99/HSE06	[138] ^e^ [106]	(2,0)-AGDYNT (6.42 Å)	0.95/PBE	[139]
(2,2)-ZGDYNT (10.25 Å)	0.65/PBE	[139]	(3,0)-AGDYNT (9.08 Å)	0.65/PBE	[139]
(3,3)-ZGDYNT (15.56 Å)	0.55/PBE	[139]	(4,0)-AGDYNT (12.04 Å)	0.55/PBE	[139]
(4,4)-ZGDYNT (20.81 Å)	0.5/PBE	[139]	AGYNRs (10–45 Å)	1.25–0.59/LDA	[140]
AGDYNRs (12–62 Å)	0.97–0.54/LDA	[140]	ZGYNRs (14–38 Å)	1.65–0.73/LDA	[140]
ZGYNRs (12–30 Å)	1.32–0.75/LDA	[140]	AGNRs (20 Å)	0.5/LDA	[141]
AGYNRs (20 Å)	0.8/LDA	[140]	ZGDYNRs (19.2–28.6 Å)	1.205–0.895/PBE	[133]
AGDYNRs (12.5–20.7 Å)	0.954–0.817/PBE	[133]	GY (−2 to +10% A-strain) 0.87–1.47	0.4–0.17/PBE 0.87–0.56/HSE06	[127]
GY (−2 to +10% H-strain)	0.4–0.88/PBE 0.87–1.47/HSE06	[127] [127]	GDY (−2 to +10% H-strain)	0.41–0.94/PBE 0.8–1.53/HSE06	[127] [127]
GY (−2 to +10% Z-strain)	0.37–0.3/PBE 0.83–0.71/HSE06	[127] [127]	GDY (−2 to +10% Z-strain)	0.39–0.21/PBE 0.78–0.56/HSE06	[127] [127]
GDY (−2 to +10% A-strain)	0.39–0.31/PBE 0.78–0.69/HSE06	[127] [127]	AGDYNRs (various n of nanoribbon)	0.04–0.69/VASP	[142]
*α*-GYs (0–10% H-strain)	0/PBE	[123]	AGDYNRs (various n of nanoribbon)	0.01–0.69/OpenMX	[142]
*β*-GYs (0–10% H-strain)	0.028–1.469/PBE	[123]	ZGDYNRs (28.6 Å)	0.895/PBE	[133]
γ-GYs (0–10% H-strain)	0.447–0.865/PBE	[123]	Graphene/0	0	[143]

^a^ Lattice constant of a = 9.44 Å and b = 6.90 Å; ^b^ lattice constants of a = 12.02 Å; ^c^ lattice constants of a = 14.6 Å; ^d^ AAA configuration presents metallic band structure; ^e^ AA stacked structure shows also metallic behavior.

**Table 3 nanomaterials-11-02268-t003:** Optimized lattice constant, interlayer distance, binding energies, and band gap for 2D and 3D graphyne and graphite ^a^.

	Lattic Constant (Å)	Interlayer Distance (Å)	Binding Energy (eV/atom)	Band Gap	[Ref.]
α	6.86	3.51	7.948	0	[136]
β1	6.86	3.27	7.963	0.19	[136]
β2	6.86	3.28	7.962	0.5	[136]
β3	6.86	3.34	7.957	0	[136]
2D GY	6.86	-	7.951	0.52	[121]
graphite	2.46	3.17	8.867	0	[136]

^a^ Using the projector-augmented wave method and the PBE functional.

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
