# Peer review of "Developments in Synthesis and Potential Electronic and Magnetic Applications of Pristine and Doped Graphynes"

_nanomaterials, 2021, doi:10.3390/nano11092268_

Round 1
Reviewer 1 Report
The manuscript is a long review (41 pages, 177 references) of the progress in graphyne science covering synthesis/growth and properties (electronic, optical, magnetic). It provides a valuable overview and collection of works. I recommend publication after the authors has considered my few suggestions for improvements below.
0) The mechanical properties and stability seems not be discussed much – maybe due to lack of works but it could be mentioned.
1) The English is improved throughout to make it more easily read (e.g. small things like grammar and use of article “the”, not using capital after period, e.g. “. graphynes”, “..a fundamental band gaps..”).
2) The structures of the various GY, GDY etc. should be introduced in a more complete way at the beginning, and the nomenclature along with it. I suggest Figure 1 and its caption is extended.
3) Page 16: The M-points in Fig 11b,c are referred to as “Dirac points”. I think the definition should be a linear, cone bandstructure around the Dirac point so a linear dispersion is clear. This is not the case for the examples in b,c.
4) In the electronic transport section it should be made more clear in the text what is theory and what is experimental. In Fig. 16 it is not clear how the gate is applied and “SCE” is not defined. Insert in B is not visible.
5) Some of the sections are very long and could be made clear by breaking them into smaller subsections, e.g. in the magnetic section page 29.
6) Fig. 20 “FC” and “ZFC” is not defined in the caption or text.
Author Response
Dear Reviewer
Thank you for your time and efforts in reviewing our article. We have carefully considered all your suggestions and modified the manuscript.
We answer your questions as follow:
- The mechanical properties and stability seems not be discussed much – maybe due to lack of works but it could be mentioned.
Thank you for this valuable suggestion. The new section (section 5, page 34) describing the mechanical properties of graphynes has been added to our review.
1) The English is improved throughout to make it more easily read (e.g. small things like grammar and use of article “the”, not using capital after period, e.g. “. graphynes”, “..a fundamental band gaps..”).
The English has been improved. All suggested changes were applied.
2) The structures of the various GY, GDY etc. should be introduced in a more complete way at the beginning, and the nomenclature along with it. I suggest Figure 1 and its caption is extended.
The complete structures of GY and GDY, in which benzene rings are linked via acetylenic and diacetylenic bridges, respectively, are shown in a new Figure 1. Furthermore, the distribution of differently hybridized carbon atoms has also been presented. The caption of Figure 2 (formerly Figure 1) has been extended.
3) Page 16: The M-points in Fig 11b,c are referred to as “Dirac points”. I think the definition should be a linear, cone bandstructure around the Dirac point so a linear dispersion is clear. This is not the case for the examples in b,c.
Thank you for pinpointing this. We have expressed in the text differences in electronic band structure occurring for α- graphyne (symmetric Dirac cones), and β- and γ-graphynes (quasi-Dirac cones).
4) In the electronic transport section it should be made more clear in the text what is theory and what is experimental. In Fig. 16 it is not clear how the gate is applied and “SCE” is not defined. Insert in B is not visible.
We have highlighted in the text which results were received in a theoretical or/and experimental way. Figure 17 (Formerly Figure 16) has been changed to make it more clear. Figure 17C has been modified to show how the gate was applied. The name of an ionic liquid used during the experiment has been introduced into the caption of Figure 17 (formerly Figure 16). The SCE is a saturated calomel electrode and has been defined in the caption of Figure 17 (Formerly 16).
5) Some of the sections are very long and could be made clear by breaking them into smaller subsections, e.g. in the magnetic section page 29.
The new subsections 3.2.1 The electronic band structure of GYs; 3.2.2 The electronic structure of GDYs; 3.3.3 The electronic band structure of GY nanoribbons and 3.3.4 The electronic band structure of bulky GYs and GDYs have been added to section 3.2. Section 4.1 has been divided into two additional sections 4.1.1 Metal-doped GYs and 4.1.2 Non-metal doped GYs and GDYs.
6) Fig. 20 “FC” and “ZFC” is not defined in the caption or text.
The FC and ZFC are abbreviations of field cooled and zero field cooled modes, respectively. The description of them has been included in the caption of Figure 21 (formerly 20).
Reviewer 2 Report
The review is written on the very urgent topic of the synthesis of graphdiyne and related structures. Due to the presence of regular distributed holes, such materials can adsorb different atoms and demonstrate significant advantages over graphene in many applications. This review provides an exhaustive description of the synthesis strategies and applications of the materials under discussion. Undoubtedly, the review will be useful for many readers, both theorists and experimenters. The high citation of this paper is very likely. In my opinion, the manuscript should be accepted in its present form.
Author Response
Dear Reviewer
Thank you for your time and efforts in reviewing our article. We are glad that the paper was met with your appreciation.
Round 2
Reviewer 1 Report
The authors has addressed all my points to my satisfaction.